Resource

EMBO
reports

# The 24-hour molecular landscape after exercise in humans reveals *MYC* is sufficient for muscle growth

Sebastian Edman [1,15], Ronald G Jones III[2,15], Paulo R Jannig[1], Rodrigo Fernandez-Gonzalo [3,4], Jessica Norrbom[5], Nicholas T Thomas [6,7], Sabin Khadgi[2], Pieter J Koopmans[2,8], Francielly Morena[2], Toby L Chambers [2], Calvin S Peterson[2], Logan N Scott[6,9,10], Nicholas P Greene[2], Vandre C Figueiredo[6,11], Christopher S Fry[6,7], Liu Zhengye[12], Johanna T Lanner[12], Yuan Wen [6,9,10], Björn Alkner[13,14], Kevin A Murach [2,8,16✉] & Ferdinand von Walden [1,16✉]

## Abstract

A detailed understanding of molecular responses to a hypertrophic stimulus in skeletal muscle leads to therapeutic advances aimed at promoting muscle mass. To decode the molecular factors regulating skeletal muscle mass, we utilized a 24-h time course of human muscle biopsies after a bout of resistance exercise. Our findings indicate: (1) the DNA methylome response at 30 min corresponds to upregulated genes at 3 h, (2) a burst of translation- and transcription-initiation factor-coding transcripts occurs between 3 and 8 h, (3) changes to global protein-coding gene expression peaks at 8 h, (4) ribosome-related genes dominate the mRNA landscape between 8 and 24 h, (5) methylation-regulated *MYC* is a highly influential transcription factor throughout recovery. To test whether MYC is sufficient for hypertrophy, we periodically pulse MYC in skeletal muscle over 4 weeks. Transient MYC increases muscle mass and fiber size in the soleus of adult mice. We present a temporally resolved resource for understanding molecular adaptations to resistance exercise in muscle (http://data.myoanalytics.com) and suggest that controlled MYC doses influence the exercise-related hypertrophic transcriptional landscape.

**Keywords** Time Course; Biopsy; Transcriptome; Methylome; Transcription Factors
**Subject Categories** Methods & Resources; Musculoskeletal System

## Introduction

Molecular alterations after a bout of exercise in skeletal muscle precede hypertrophic adaptation and ultimately contribute to a change in phenotype (Egan et al, 2013; Egan and Sharples, 2022; Gustafsson et al, 1999; Jozsi et al, 2000; Kraniou et al, 2000; Perry et al, 2010; Pilegaard et al, 2000). Initial time course work in humans that used ≥3 post-exercise muscle biopsies established 2-4 h into recovery as the ideal time point for studying targeted changes in mRNA levels after a bout of exercise (Louis et al, 2007; Pilegaard et al, 2003; Vissing et al, 2005; Yang et al, 2005). Others that leveraged more comprehensive profiling of global gene expression (Mahoney et al, 2005; Neubauer et al, 2014; Zambon et al, 2003) demonstrated that many genes have delayed and/or biphasic responses to exercise in muscle that extend beyond 4 h. Recent work in skeletal muscle further emphasizes that gene expression data from a single time point after exercise is limiting when trying to capture the complex and dynamic nature of the adaptive response, and could even lead to inaccurate or misleading conclusions (Kuang et al, 2022). It is also important to consider the effects of the muscle biopsy (Vissing et al, 2005) and circadian rhythm (Zambon et al, 2003) in human exercise studies; these factors are typically overlooked. There is a critical need for temporally resolved and biopsy-only controlled investigations to explore and understand the molecular responses to resistance exercise since muscle mass and function is strongly associated with all-cause mortality (Isoyama et al, 2014; Li et al, 2018; Metter et al, 2002; Newman et al, 2006). A detailed understanding of the most influential molecular factors during the post-resistance exercise recovery period will help focus efforts at developing targeted therapies against muscle mass loss and/or enhancing hypertrophic responsiveness to exercise interventions.

[1]Division of Pediatric Neurology, Department of Women's and Children's Health, Karolinska Institute, Stockholm, Sweden. [2]Exercise Science Research Center, Department of Health, Human Performance and Recreation, University of Arkansas, Fayetteville, AR, USA. [3]Division of Clinical Physiology, Department of Laboratory Medicine, Karolinska Institute, Stockholm, Sweden. [4]Unit of Clinical Physiology, Karolinska University Hospital, Huddinge, Sweden. [5]Molecular Exercise Physiology Group, Department of Physiology and Pharmacology, Karolinska Institute, Stockholm, Sweden. [6]Center for Muscle Biology, University of Kentucky, Lexington, KY, USA. [7]Department of Athletic Training and Clinical Nutrition, University of Kentucky, Lexington, KY, USA. [8]Cell and Molecular Biology Graduate Program, University of Arkansas, Fayetteville, AR, USA. [9]Department of Physiology, University of Kentucky, Lexington, KY, USA. [10]Division of Biomedical Informatics, Department of Internal Medicine, University of Kentucky, Lexington, KY, USA. [11]Department of Biological Sciences, Oakland University, Rochester Hills, MI, USA. [12]Molecular Muscle Physiology & Pathophysiology Group, Department of Physiology & Pharmacology, Karolinska Institute, Stockholm, Sweden. [13]Department of Orthopaedic Surgery, Region Jönköping County, Eksjö, Sweden. [14]Department of Biomedical and Clinical Sciences, Linköping University, Linköping, Sweden. [15]These authors contributed equally as first authors: Sebastian Edman, Ronald G Jones III. [16]These authors contributed equally as senior authors: Kevin A Murach, Ferdinand von Walden. ✉E-mail: kmurach@uark.edu; ferdinand.von.walden@ki.se

Several seminal (Alway, 1997; Armstrong and Esser, 2005; Chen et al, 2002) and recent studies (Murach et al, 2022; Viggars et al, 2022a) suggest that the transcription factor *c-Myc* (referred to as *Myc* or *MYC* for mouse and human genes, respectively) is a key component of skeletal muscle hypertrophic adaptation to loading in animals. Our work using human skeletal muscle biopsies after a bout of resistance exercise (RE) (Figueiredo et al, 2021), as well as meta-analytical information that combines numerous human muscle gene expression datasets during the recovery period after exercise (Pillon et al, 2020), indicates that *MYC* is highly responsive to hypertrophic loading (Jones et al, 2022). MYC protein accumulates in human muscle following a bout of RE (Broholm et al, 2011; Brook et al, 2016; Figueiredo et al, 2016; Townsend et al, 2016) as well as in response to chronic training (Stec et al, 2016). Its expression may also differentiate between low and high hypertrophic responders (Stec et al, 2016). *Myc* is induced cell-autonomously in myotubes by electrical stimulation in vitro (Sidorenko et al, 2018) and is strongly upregulated in murine myonuclei during mechanical overload (Murach et al, 2022). MYC protein localizes to myonuclei during loading-induced hypertrophy (Alway, 1997; Armstrong and Esser, 2005), is considered pro-anabolic (Dang, 1999, 2012; Das et al, 2022), and can drive muscle protein synthesis and ribosome biogenesis in skeletal muscle (Brook et al, 2016; Mori et al, 2020; Wen et al, 2016; West et al, 2016). Loss of MYC results in lower muscle mass in preclinical models (Demontis and Perrimon, 2009; Wang et al, 2023). MYC is also estimated to target ~15% of the genome across different tissues and species (Dang et al, 2006). Still, the magnitude of its contribution to the exercise response in humans is not entirely understood (Phillips et al, 2013), nor is its sufficiency for muscle hypertrophy in preclinical models.

The current investigation details the global gene expression response to a bout of RE after 30 min, 3-, 8-, and 24-h using RNA-sequencing (RNA-seq) in skeletal muscle biopsy samples from healthy untrained humans. We reveal the effect of the muscle biopsy and inherent circadian rhythmicity using a biopsy-only, feeding and time point-matched control group. The human transcriptional time course data is provided in a publicly available user-friendly web-based application at http://data.myoanalytics.com. We then analyzed the human muscle methylome at 30 min after RE and combined these data with the transcriptome response to RE using a novel -omics integration approach. Integration of methylomics and transcriptomics sheds light on the molecular regulation of gene expression during recovery from exercise. With our transcriptome data, we infer the major transcriptional regulators of the exercise response using in silico ChIP-seq (Qin et al, 2020) that we have previously validated in skeletal muscle with a genetically modified mouse (Jones et al, 2022; Murach et al, 2022). These molecular and computational analyses identified *MYC* as an influential transcription factor controlling the exercise transcriptome throughout the time course of recovery after a bout of RE. Muscle-specific *Myc* overexpression data from the plantaris (Murach et al, 2022) and soleus (Jones et al, 2022) of mice reinforced the human exercise data. We employed a genetically modified mouse model to induce MYC in a pulsatile fashion specifically in skeletal muscle over 4 weeks to determine if MYC is sufficient for hypertrophy. Our genetically driven pulsatile approach avoids potential negative effects of chronically overexpressing a hypertrophic regulator (Castets et al, 2013; Ham et al, 2020) and more closely mimics the transient molecular response of exercise in skeletal muscle (Egan et al, 2013;

Egan and Sharples, 2022; Perry et al, 2010; Smith et al, 2023). This work collectively illustrates the molecular landscape with temporal resolution after a bout of RE and places *MYC* at the center of the skeletal muscle RE response in mice and humans.

# Results

## Biopsy time course at rest and the transcriptional regulation of circadian genes in human skeletal muscle

For the analysis of RNA-seq data, we focused on protein-coding genes as they are the most well-characterized. The results of these mRNAs are presented in Dataset EV1 and EV2. The Pre muscle biopsies were taken 15 min prior to the 45-min control protocol (equivalent to the 45 min of resistance exercise in RE group; Figs. 1A and 2A). Thus, in practice, the biopsy named 30 min post is taken 90 min after the Pre biopsy (15 min plus, 45 min, plus 30 min), the biopsy named 3 h is taken 4 h after the Pre biopsy, and so on. The biopsy time points, and their naming are chosen to precisely correspond to the post RE time points for the RE group (Fig. 2A). The first muscle biopsies were obtained within a 3.5 h window beginning at 7:30 AM for all participants in the study.

Differentially expressed genes (DEGs, adj. $p < 0.05$) were analyzed relative to the collected Pre biopsy. In the control group (CON, $n = 5$) the number of DEGs at the different recovery time points was: 30 min—0 upregulated, 1 downregulated; 3 h—12 upregulated, 30 downregulated; 8 h—55 upregulated, 75 down-regulated; and 24 h—0 upregulated, 1 downregulated (Fig. 1B,D). Thus, the most protein-coding DEGs were observed at 8 h relative to Pre. Previous work involving human skeletal muscle biopsies revealed the rhythmic expression of muscle circadian core clock genes over 24 h (Perrin et al, 2018). We found that *NR1D1* (*REVERBα*) (adj. $p = 0.01 \times 10^{-5}$), *PER1* (adj. $p = 0.0008$), and *PER2* (adj. $p = 0.007 \times 10^{-6}$) were lower at the 3-h time point (Fig. 1C). At 8 h, *NR1D2* (*REVERBβ*) (adj. $p = 0.005$), *PER1* (adj. $p = 0.03 \times 10^{-6}$), *PER2* (adj. $p = 0.007 \times 10^{-10}$), and *PER3* (adj. $p = 0.007 \times 10^{-3}$) were lower, while *ARNTL*, also known as *BMAL1*, was upregulated (adj. $p = 0.01$) (Fig. 1C). *KLF15*, a circadian-regulated mediator of lipid metabolism (Perrin et al, 2018), was lower at the 3-h (adj. $p = 0.0001$) and 8-h time points (adj. $p = 0.03 \times 10^{-12}$) in the CON group (Fig. 1C). *PPARGC1β*, another circadian-controlled gene (Gidlund et al, 2015; McCarthy et al, 2007), was upregulated at 3 h (adj. $p = 0.048$) and 8 h (adj. $p = 0.01$; Fig. 1C). Also worth mentioning is that *FOXO3*, a central regulator of autophagy and mass in skeletal muscle (Mammucari et al, 2007; Sandri et al, 2006; Zhao et al, 2007), was lower at 3 h (adj. $p = 0.0004$) and 8 h (adj. $p = 0.007 \times 10^{-3}$, Fig. 1C). *SESN1*, which may also regulate muscle mass (Segalés et al, 2020), was lower at 3 h (adj. $p = 0.003$) and 8 h (adj. $p = 0.008 \times 10^{-3}$) (Fig. 1C).

To more broadly investigate what specific functions were being regulated as a consequence of our CON intervention, we ran background corrected gene set enrichment analysis using Enrichr with the 2023 gene ontology (GO) database as our cross reference (GO: biological process & molecular function) and DEGs combined across all time points (Aleksander et al, 2023; Chen et al, 2013; Kuleshov et al, 2016; Stokes et al, 2023; Xie et al, 2021). Collectively, no significantly upregulated gene sets were detected (Fig. 1E). A few negatively regulated GO-based gene sets were detected, indicating a

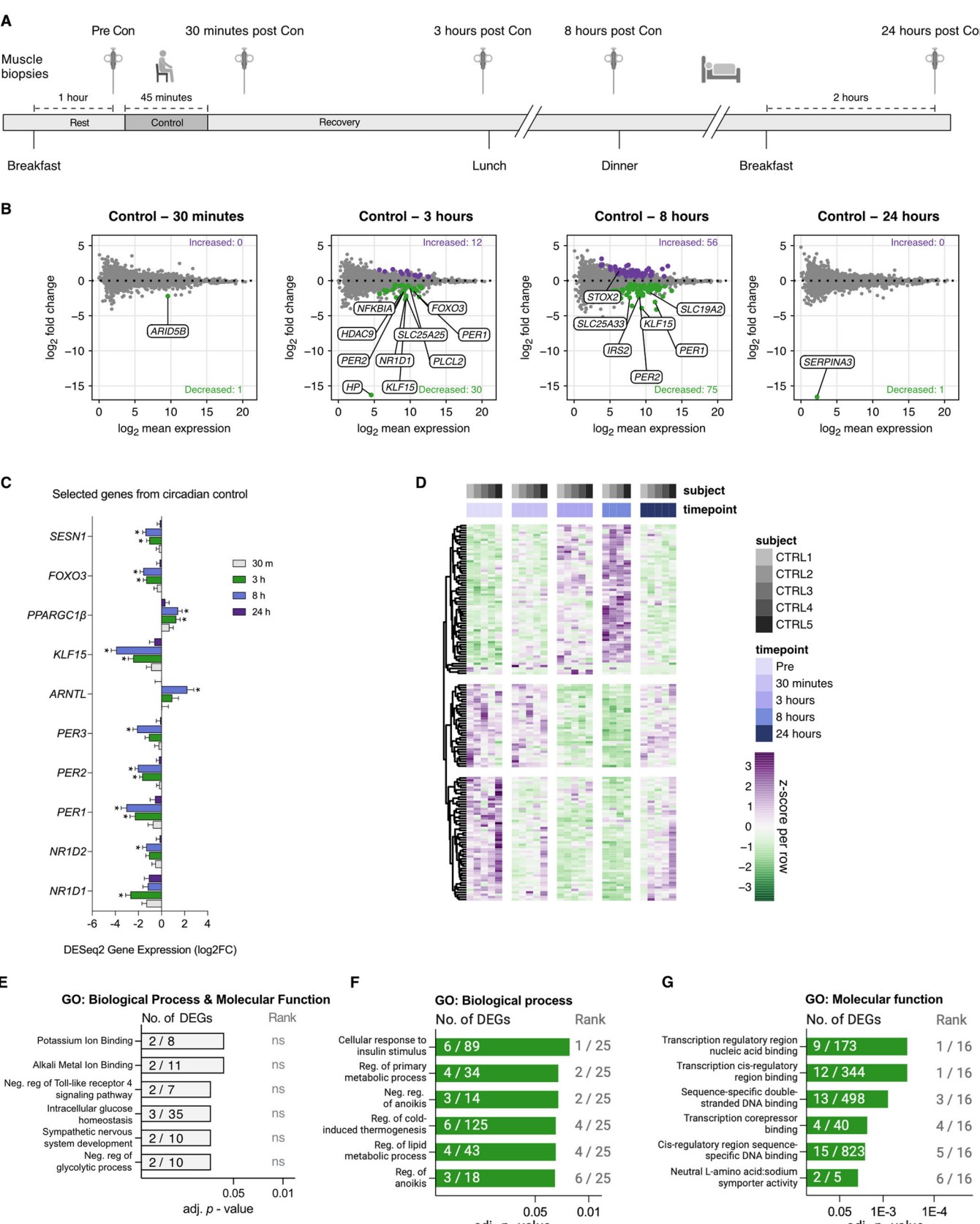

**Figure 1.  Gene expression patterns for biopsy-only control time course.**

(A) Schematic overview for the control arm of the human intervention, $n = 5$. (B) MA plots showing differentially expressed genes (DEG) vs pre-values, time matched to 30 min, 3, 8, & 24 h of recovery in the resistance exercise trial (see Fig. 2A). Purple and green dots indicate up- or downregulated regulated genes (adj. $p < 0.05$), respectively. Top genes for adj. $p$-value are highlighted in plots. (C) Fold-change for targeted DEGs in the control group, $n = 5$. *adj. $p < 0.05$. Values represent $\log_2$ fold-change ± SEM. *SESN1* $p = 0.0034$ at 3 h, $p = 8.7E{-}6$ at 8 h, FOXO3 adj. $p = 0.0004$ at 3 h, adj. $p = 7.7E{-}6$ at 8 h, PPARGC1B adj. $p = 0.0486$ at 3 h, adj. $p = 0.0110$ at 8 h, *KLF5* adj. $p = 1.0E{-}5$ at 3 h, adj. $p = 3.5E{-}14$ at 8 h, *ARNTL* adj. $p = 0.0120$ at 8 h, *PER3* adj. $p = 7.8E{-}6$ at 8 h, *PER2* adj. $p = 6.8E{-}9$ at 3 h, adj. $p = 7.3E{-}13$ at 8 h, *PER1* adj. $p = 8.3E{-}5$ at 3 h, adj. $p = 3.0E{-}8$ at 8 h, *NR1D2* adj. $p = 0.0058$ at 8 h, *NR1D1* $p = 1.1E{-}7$ at 3 h. (D) Heatmap showing z-scores for 60 up-, and 90 downregulated DEGs across all time points and volunteers in the control trial. (E, F) Gene ontology (GO) gene set enrichment analysis on DEGs across the entire 24-h control period. Numbers within the bars indicate the proportion of DEGs in our dataset corresponding to the specific gene set. Rank values indicate the specific gene sets adj. $p$-value rank. (E) Upregulated Biological Processes and Molecular Functions, (F) Downregulated Biological Functions, (G) Downregulated Molecular Functions. (B, C) DESeq2 was calculated using a Wald test with a Benjamini–Hochberg $p$-value correction. (E–G) Gene ontology (GO) gene set enrichment analysis is analyzed using a Fisher exact test with Benjamini–Hochberg $p$-value correction. Con control situation, ns not significant, Neg. negative, Reg. regulation. Source data are available online for this figure.

reduced "cellular response to insulin" and reduced "transcriptional speed" (Fig. 1F,G). Although significantly enriched, only 2–15 genes were coding for each gene set, suggesting the effect was small. Nevertheless, it should be noted that the significantly downregulated gene sets are almost exclusively driven by DEGs expressed at 3 and 8 h. In addition to circadian rhythmicity of gene expression, DEGs across time points in CON could be related to feeding since the Pre, 30-min, and 24-h biopsies were taken 1, 2, and 2 h after feeding, respectively, while the biopsies at 3 and 8 h were both taken approximately 5 h after food intake (Fig. 1A). Thus, our finding of slightly downregulated insulin stimulation response and transcription genes at 3 and 8 h seems intuitive.

## Differentially expressed genes peaked 8 h after resistance exercise (RE) relative to pre

Following an acute bout of resistance exercise (RE, $n = 8$, Fig. 2A), the number of DEGs relative to the Pre time point (adj. $p < 0.05$) was: 30 min—64 upregulated, 5 downregulated; 3 h—1281 upregulated, 1298 downregulated; 8 h—1764 upregulated, 1253 downregulated; and 24 h—751 upregulated, 445 downregulated (Fig. 2B). Over 90% of the transcriptomic signatures were estimated to originate from myofibers irrespective of time point. This proportion was estimated using CIBERSORTx (Newman et al, 2019), a computational cellular devonvolution tool that we have used previously for inferring changes in cell type after an acute hypertrophic stimulus (Murach et al, 2022). A recent acute exercise and single-cell RNA-seq study in human skeletal muscle was used as the reference dataset (Lovrić et al, 2022). The exercise involved intense cycle sprint intervals and the muscle sampled was the vastus lateralis, which corresponds with our study design. In this dataset (Lovrić et al, 2022), "myocytes" were inferred to be myonuclei based on the expression of adult myosin heavy chains and other muscle fiber-enriched markers. Myonuclei usually appear in skeletal muscle single-cell datasets and are sequenced alongside mononuclear cells (McKellar et al, 2021; Murach et al, 2021c). When excluding "myocytes" (myofibers), no appreciable change in mononucleated cell proportions was estimated throughout recovery (Fig. EV1). As anticipated, we noted changes in genes previously recognized as responsive to RE and/or important for muscle remodeling (Fig. 2C) (Correia et al, 2023; Ferreira et al, 2019; Figueiredo et al, 2021; Pillon et al, 2020). Of all upregulated protein-coding genes across the 24-h recovery period in the RE group, 46% were differentially expressed at two or more time points while the proportion was 34% for downregulated genes. In total,

2399 unique upregulated and 2126 unique downregulated DEGs were identified throughout the 24-h recovery period (Fig. 2D). DEGs at each time point relative to Pre are presented in Dataset EV2.

## The integrated 24-h recovery transcriptome after acute RE

Using the two lists of all DEGs from across the entire 24-h recovery period after RE (up- or downregulated) relative to Pre generated in Fig. 2D (Complete list; Dataset EV2), we employed background corrected gene set analysis as described above on each list separately (Aleksander et al, 2023; Chen et al, 2013; Kuleshov et al, 2016; Stokes et al, 2023; Xie et al, 2021).

For the 2399 upregulated protein-coding DEGs across the entire time course of recovery, 44 biological processes (adj. $p < 0.05$) were identified, with a large proportion of the gene sets in the 24-h post-RE window relating to transcription, translation, and the synthesis of new ribosomes. After exclusion of gene sets with large overlaps in underlying DEGs, the top (adj. $p$-value ranked) processes were ribosome biogenesis (GO:0042254), activation of protein localization to telomere (GO:1904816), inhibition of apoptosis (GO:0043066), activation of transcription by RNA polymerase II (GO:0045944), activation of intracellular signal transduction (GO:1902533), and response to unfolded protein (GO:00066986) (Fig. 3A). Fourteen molecular function gene sets were also identified (adj. $p < 0.05$). Of the molecular functions identified, RNA binding (GO:0003723), cadherin binding (GO:0045296), ubiquitin protein ligase binding (GO:0031625), purine ribonucleoside triphosphate binding (GO:0035639), protein phosphatase 2A binding (GO:0051721), and MAP kinase tyrosine/serine/threonine phosphatase activity (GO:0033550) were the top gene sets, again excluding gene sets with large overlap (Fig. 3B).

Next, we identified at which time point all DEGs within each specific gene set were differentially upregulated relative to Pre. Using the gene set analysis from the entire time course of recovery, the number of upregulated DEGs in a gene set at each specific time point was expressed as a percentage of the entire gene set response (e.g., 61 DEGs in our dataset were found to regulate ribosome biogenesis, and of these 61 genes, 57—or 93%—were enriched following 8 h of recovery). Plotting these values for each individual time point relative to Pre thus revealed a 24-h temporal pattern of each gene set following acute RE (Fig. 3D,E). None of the most highly enriched gene sets within our analysis peaked at 30 min post-exercise. However, a targeted analysis of enriched gene sets

with peak expression at 30 min revealed growth factor- and glucocorticoid responses are strongest at 30 min post-RE, as well as stress response signaling and mRNA catabolism (Fig. EV2A). The peak in transcripts coding for mRNA catabolism 30 min after RE (*BTG2*; adj. $p = 0.005$, *ZC3H12A*; adj. $p = 0.003$, *ZFP36L1*; adj. $p = 0.01$, and *TOB1*; adj. $p = 0.0005$) precedes any marked down-regulation of DEGs, suggesting catabolism of mRNA occurs in muscle following upregulation of anti-proliferative- and mRNA-decaying enzymes.

At 3 h post-RE relative to Pre, we observed two major upregulated gene sets that were peaking: response to unfolded proteins (20/44 genes; Fig. 3A,D) and MAP kinase phosphatase activity (6/8 genes; Fig. 3B,E). The former of the two is primarily driven by genes coding the heat shock protein family, such as *DNAJA1* (adj. $p = 8.5 \times 10^{-7}$ at 3 h) and *HSPA1A* (adj. $p = 8.5 \times 10^{-5}$ at 3 h). The latter, MAP kinase phosphatase activity, is driven by genes encoding the dual specificity phosphatase protein family (DUSP), responsible for dephosphorylation of tyrosine/serine/threonine sites (*DUSP2, 4, 5, 8, 10 & 16*, adj. $p < 0.05$ at 3 h). DEGs within this gene set peaked at the 3-h time point, with only two remaining elevated at 8 h. This pattern was also reflected when mapping the fold change of DEGs within each gene set, rather than the number of DEGs, across the 24-h recovery (Fig. 3G,H). A clear peak in genes encoding phosphatase activity directed towards the MAP kinase superfamily early during RE recovery may be a response triggered by the rapid severalfold increase in protein phosphorylation of mTOR-targets such as S6K1 and 4EBP1 occurring at around 60–90 min post-RE (Apró et al, 2015; Moberg et al, 2016). According to this previous work, the rapid rise in anabolic signaling via protein phosphorylation at this time point is then followed by a swift decrease, with some signaling proteins showing close to baseline phosphorylation levels at 3 h of recovery (Apró et al, 2015; Moberg et al, 2016).

Several upregulated gene sets are overrepresented to a similar degree at 3 and 8 h of recovery such as ubiquitin protein ligase binding (70/259 genes; GO:0031625; Fig. 3B,E,H), activation of transcription by RNA polymerase II (171/745 genes; GO:0045944; Fig. 3A,D,G), and activation of intracellular signal transduction (111/446 genes; GO:1902533; Fig. 3A,D,G), pointing to increased protein turnover. At the same time, inhibition of apoptosis (104/385 genes; GO:0043066; Fig. 3A,D,G) was also upregulated, which may be a direct response to the increased transcriptional emphasis on the ubiquitin system. The post-translational modifications mediated by ubiquitination of pro-apoptotic Bcl-2 family and BH3-only proteins have been proposed to be crucial for cell survival (Roberts et al, 2022). For instance, the transcript *RNF144B* coding for the E3 ubiquitin ligase *IBRDC2*, which targets the Bcl-2 'executioner' Bax for ubiquitination (Benard et al, 2010), is significantly upregulated at 3 h only (adj. $p = 2.8 \times 10^{-7}$).

Following the burst of transcription- and translation initiation-coding transcripts at 3 and 8 h, the upregulated mRNA landscape shifted toward the ribosome. At 8 and 24 h of recovery, regulation of ribosome biogenesis (61/151 genes; GO:0042254; Fig. 3A,D,G) and RNA binding (340/1289 genes; GO:0003723; Fig. 3B,E,H) appear to be the dominant gene sets. Within the RNA binding gene set, genes supporting ribosome assembly, posttranslational control of RNA, splicing via RNA-binding motif protein family members (e.g., *RBMX*, *RBM15*, *RBM39*), heterogeneous nuclear ribonucleo-proteins (e.g., *HNRNPU*, *HNRNPR*, *HNRNPC*) and zinc fingers

(e.g., *ZNF326*, *ZNF579*, *ZNF697*) were differentially expressed. Moreover, several transcripts coding ribosome biogenesis factors (e.g., *BMS1* and *LTV1*) as well as ribosomal assembly and transport proteins (e.g., *NIP7*, *NOP14*, *RPF2*) comprised the highly enriched ribosome biogenesis gene set (GO:0042254).

Gene sets enriched within the 2126 downregulated DEGs were considerably fewer compared to the upregulated genes (Fig. 3C,F,I). Here, five biological processes and 13 molecular functions were downregulated (adj. $p < 0.05$). Out of the five significant biological processes, four were related to transcription. The majority of DEGs within these transcription-related gene sets were classified as inhibitors of transcription, meaning RE likely acts on transcription by up-regulating activation (Fig. 3A) and by repressing suppressor genes (Fig. 3C) to a similar extent. In addition to transcription, histone H3 methyltransferase activity was one of the gene sets found to be significantly downregulated at 3 and 8 h following RE using a targeted analysis of these specific time points (Fig. EV2B).

## Information on genes with divergent responses throughout the time course of RE recovery

Specifically focusing on genes that were upregulated early after RE and downregulated later relative to Pre (Cluster 4; Fig. 2D), we found that some of the negative regulators of RNA Pol II transcription (GO:0045892, GO:0000122) followed this pattern—upregulated 30 min and/or 3 h while downregulated later at 8 and 24 h (Fig. EV2C). Nuclear receptor subfamily 4 group A genes *NR4A1* and *NR4A2* were among the 10 transcripts in the sequence-specific DNA binding gene set (GO:0043565) that most clearly followed a biphasic expression pattern (up early, down late; Fig. EV2D). Related to *NR4A1* and *NR4A2*, *NR4A3* showed similar biphasic tendencies, being upregulated at 3 h (4.93 log$_2$FC, adj. $p < 0.05$) and shifting toward lower expression at 24 h relative to Pre (−0.78 log$_2$FC, adj. $p > 0.05$; Fig. EV2D). This finding is in agreement with previous reports suggesting *NR4A* family transcripts are highly responsive early during exercise recovery (Amar et al, 2021; Pillon et al, 2020). Others have also shown that *NR4A3* is upregulated by seemingly opposing stimuli—following both acute exercise as well as long-term inactivity (Amar et al, 2021; Pillon et al, 2020). Many acute exercise interventions only sample muscle for 3–5 h during recovery, so a delayed depression of the *NR4A* genes (at 24 h into recovery) due to the biphasic nature of this gene family (Fig. EV2D) may be underappreciated. An interpretation could be that exercise ubiquitously drives *NR4A* expression, while in fact, the post-exercise induction and repression could be balanced, and this pattern may have a specific biological function pertaining to exercise adaptation. Regardless, the existence of biphasic genes within 24 h of RE recovery highlights the importance of considering muscle biopsy sampling time points when interpreting data.

## Harmonizing the biopsy-only time course with the RE recovery time course

The biopsy-only group that did not undergo exercise provides a lens into the effects of circadian rhythmicity, diet, and/or the effect of the muscle biopsy and how this relates to the RE response. Out of the 60 upregulated, and 90 downregulated genes expressed across the 24-h recovery in the biopsy-only group, 28 and 58 genes respectively were

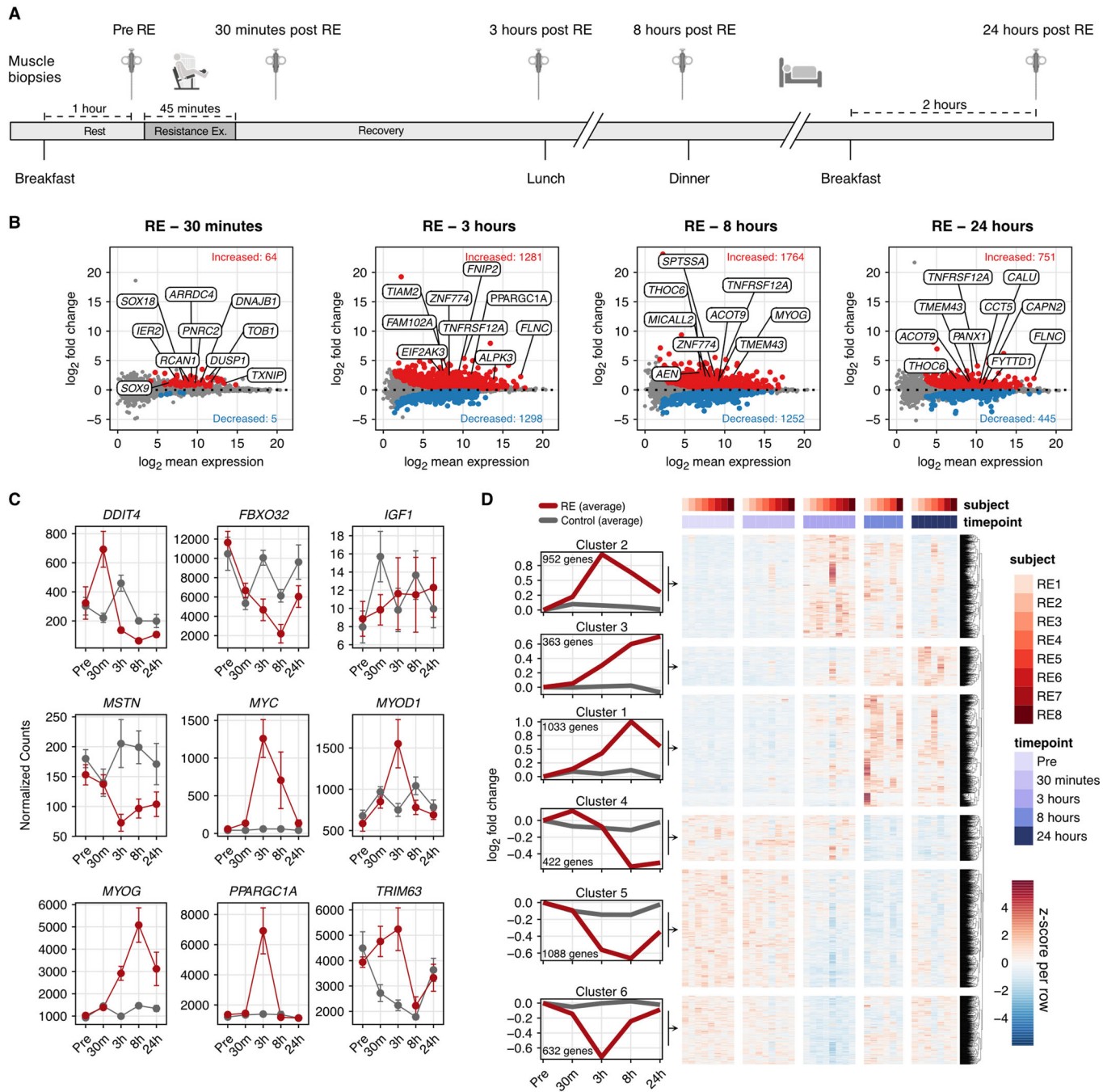

**Figure 2. Gene expression patterns during 24 h of recovery from resistance exercise.**

(A) Schematic overview of resistance exercise (RE) intervention, *n* = 8. (B) MA plots showing differentially expressed genes (DEG) vs resting pre-values, following 30 min, 3, 8, & 24 h of recovery from RE (adj. *p* < 0.05). DESeq2 was calculated using a Wald test with a Benjamini–Hochberg *p*-value correction. Red and blue dots indicate up- or downregulated regulated genes (adj. *p* < 0.05), respectively. Top 10 genes for adj. *p*-value are highlighted in plots. (C) Normalized counts for targeted genes across the 24-h intervention, *n* = 8. Values represent normalized counts ± SEM. Red dots = RE trial, gray dots = Control trial. (D) Heatmap showing z-scores for 2399 up-, and 2126 downregulated DEGs across all recovery time points and volunteers in the RE trial. Genes are clustered according to their expression pattern across time points within the 24-h recovery period. (B–D) DESeq2 was calculated using a Wald test with a Benjamini–Hochberg *p*-value correction.

similarly regulated in the RE group (Fig. EV3). Among them were circadian genes *PER1* and *PER3*, but not *PER2* or *PPARGC1β* (presented in Fig. 1C). By contrast, 8 upregulated DEGs in the control group were downregulated by RE, and 9 genes that were downregulated in the control group were instead upregulated by RE. Of the genes that overlapped between the biopsy-only group and the RE group (e.g., *KLF15, NR1D1, NR1D2, PER1, SESN1*), RE tended to blunt their excursions and increase variability. For example, at the 8-h

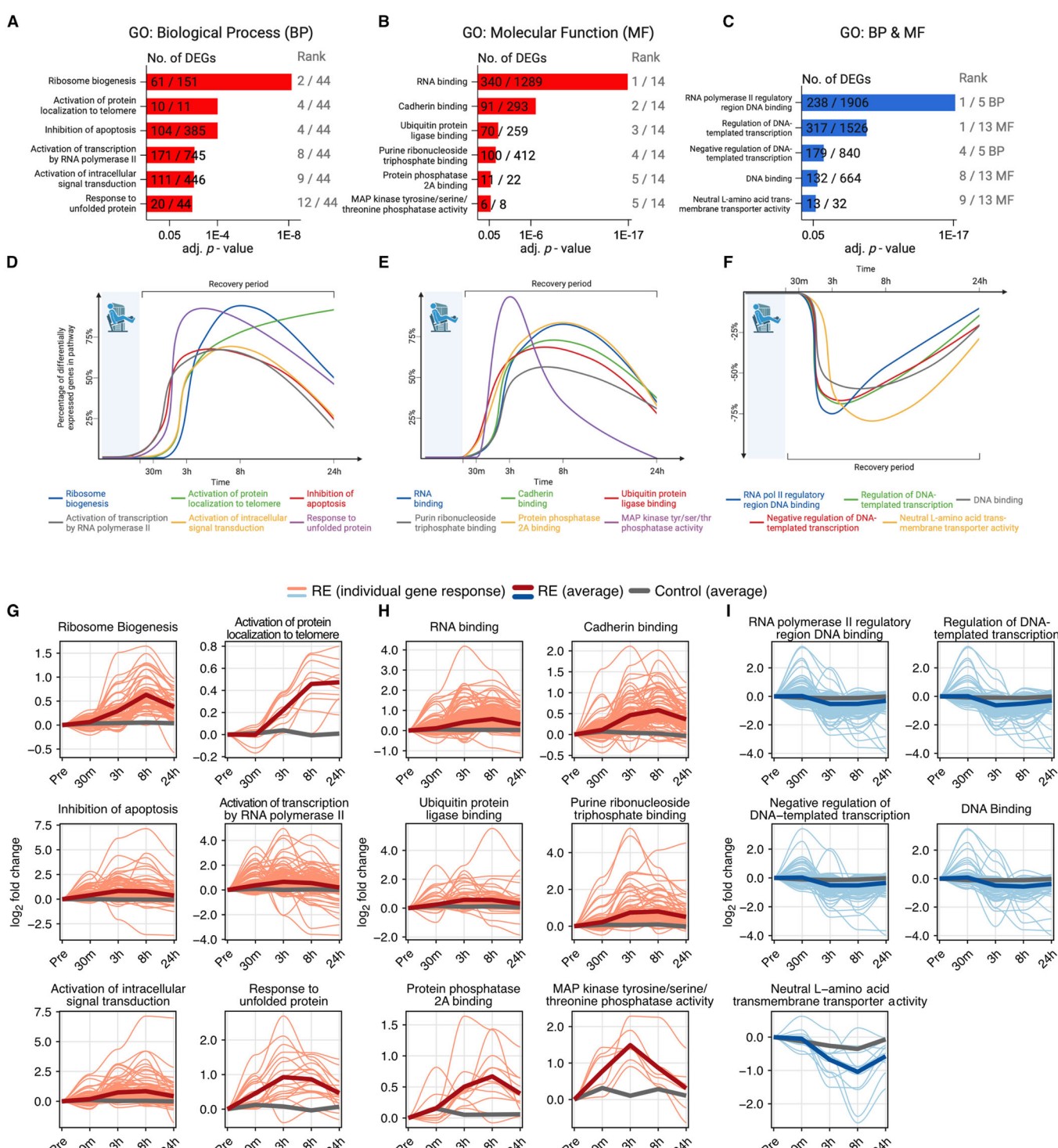

time point, *KLF15* was −3.9 log₂FC lower (adj. $p = 0.04 \times 10^{-12}$) relative to Pre in the biopsy-only group, whereas after RE at the same time point, *KLF15* was −1.9 log₂FC lower relative to Pre (adj. $p = 0.02 \times 10^{-4}$) with a larger range of expression. As expected, the effect of RE outweigh any other effects with respect to overall gene expression, but there appears to be interaction between exercise and the biopsy, feeding, and or circadian rhythmicity across time points that should be considered when designing muscle biopsy studies.

## Inferring fiber type distribution from the transcriptional data across cohorts

To characterize the fiber type distribution of the RE and CON group, we used the gene counts of adult myosin heavy chain mRNAs (*MYH1*—Type IIX, *MYH2*—Type IIA, and *MYH7*—Type I) to interpolate skeletal muscle fiber type distribution. We accomplished this by leveraging a publicly available dataset that contained both

**Figure 3.   Gene set enrichment time course across 24 h of recovery from resistance exercise.**

(A–C) Gene ontology (GO) gene set enrichment analysis on DEGs across the entire 24-h recovery period. Numbers within the bars indicate the proportion of DEGs in our dataset corresponding to the specific gene set. Rank values indicate the specific gene sets adj. p-value rank. The corresponding timeline shows the proportion of DEGs with the specific gene sets across the 24-h recovery period. (A) Upregulated Biological Processes, (B) Upregulated Molecular function, and (C) Downregulated Biological Processes and Molecular function. (D–F) Timelines of the top (GO) gene sets from (A–C) expressed as a percentage of the number of DEGs within that gene set. (D) Upregulated Biological Processes, (E) Upregulated Molecular function, and (F) Downregulated Biological Processes and Molecular function. (G–I) Average fold-change (vs Pre) for highlighted gene sets (thick red lines) along individual DEGs within the specific gene set (lighter red lines). The average fold-change for the same genes in the control situation is presented in gray. (G) Upregulated Biological Processes, (H) Upregulated Molecular function, (I) Downregulated Biological Processes and Molecular Function. (A–C) Gene ontology (GO) gene set enrichment analysis is analyzed using a Fisher exact test with Benjamini–Hochberg p-value correction. Source data are available online for this figure.

transcriptional data and fiber type distribution using muscle histology (Reitzner et al, 2024). Correlating the % Type I fiber area from each individual to the MYH transcript ratio ($MYH7/(MYH2 + MYH1)$) yielded a strong significant correlation of $r = 0.7178$ ($p < 0.0001$, Fig. EV4A). We then used these values to create a standard curve and inserted our *MYH*-transcript data along the fitted line, thus estimating a fiber-type distribution in our samples. The muscle samples from the CON group and RE group consisted of $54.3 \pm 9.2$ and $60.5 \pm 12.2\%$ type I fiber area (Fig. EV4B), respectively, with no difference between the two groups ($p = 0.33$).

## Changes to the muscle DNA methylome at 30 min of recovery after RE relates to mRNA responses at 3 h

Binding and expression target analysis (BETA) is a multi-omics integration method for understanding transcriptional regulation (Wang et al, 2013). We recently adapted this method for combining reduced representation bisulfite sequencing (RRBS) data with RNA-sequencing data to understand how DNA methylation regulates the transcriptome during an acute loading stimulus in mice (Ismaeel et al, 2023). We also used this technique to relate the methylome to the proteome after exercise training in skeletal muscle (Chambers et al, 2024). Briefly, BETA considers differential methylation status (both hypo- and hyper-methylation) in relation to transcription start sites using weighted scores to infer transcriptional regulation, which is then combined with transcriptomic data for validation. This method generates a regulatory potential score on a gene-by-gene basis as well as an overall p value for a cumulative distribution function (one-tailed Kolmogorov–Smirnov test) that discriminates global time point differences for up or down genes. In our recent work, myonuclear DNA methylation status coincided with changes in myonuclear gene expression as well as the acute metabolic responses that occurred during rapid muscle growth, giving us confidence in the validity of BETA (Ismaeel et al, 2023). We leveraged RRBS and RNA-sequencing data in the current study to provide a deeper understanding of transcriptional regulation in response to acute RE.

We first used BETA to compare the methylome and transcriptome responses to RE at 30 min of recovery versus Pre (Dataset EV3 shows processed methylation data for 30 min post-RE versus Pre). Combining datasets at this time point revealed <10 genes were likely being regulated at the level of methylation. This result seems intuitive since changes in DNA methylation typically precede changes in gene expression (Barres et al, 2012), which does not peak until later time points in our data. As such, we combined the 30 min methylome data with the later transcriptome time

points after RE. Changes to the methylome 30 min after RE were strongly predictive of the changes observed in gene expression at 3 h after RE versus Pre (Fig. 4A), but not later time points. This analysis inferred significant methylation control of 936 upregulated genes at 3 h ($p = 0.000007$), and 805 downregulated genes were identified according to BETA ($p < 0.05$), but the overall regulation of repressed genes was not significant according to the Kolmogorov–Smirnov test ($p = 0.952$). It is important to note that the lack of significance according to BETA for downregulated genes does not mean that methylation is not regulating gene expression on a gene-by-gene basis, but that the global regulatory potential score did not achieve significance. Thus, we present the BETA analysis for individual genes to provide additional insights.

Of upregulated genes with a coordinated methylome and transcriptome response, *TNFRSF12A* (*FN14*) was the most significant ($p = 0.000035$; Fig. 4B). Upregulation of the TWEAK receptor *FN14* occurs during the muscle hypertrophic response to exercise specifically in fast-twitch type 2 fibers of humans (Murach et al, 2014; Raue et al, 2012). This role for *FN14* induction during muscle adaptation could be related to non-canonical NF-$\kappa$B pathway activation (Raue et al, 2015). Furthermore, inhibition of *FN14* in human myotubes increases C/EP$\beta$ and MuRF (Walton et al, 2019). Alternatively, mechanistic work in rodents suggests *Fn14* knockout in muscle fibers improves endurance exercise capacity and inhibits neurogenic muscle atrophy (Tomaz da Silva et al, 2022), but ablation in satellite cells attenuates muscle regeneration (da Silva et al, 2023). More gain and loss of function studies are needed to clarify the role of *Fn14* in hypertrophic muscle adaptation (Dungan et al, 2022; Pascoe et al, 2020). Other notable genes with a coordinated upregulated response to RE included: *RUNX1* (Fig. 4B), which regulates muscle mass (Wang et al, 2005) and is enriched in myonuclei during rapid load-induced hypertrophy (Murach et al, 2022); *RBM10* (Fig. 4B), an RNA splicing factor that we previously showed is altered at the methylation level in muscle with late-life hypertrophic exercise in mice (Dungan et al, 2022; Murach et al, 2021a); and *NR4A3* (Fig. 4B), among the most exercise-responsive genes in skeletal muscle that controls metabolism (Pillon et al, 2020). We previously reported that promoter region CpG hypomethylation of *Myc* in myonuclei (von Walden et al, 2020) coincided with strong upregulation of myonuclear and muscle tissue *Myc* levels during acute mechanical overload in mice (Murach et al, 2022; von Walden et al, 2020). BETA also suggested coordinated methylation and transcriptional regulation of *MYC* by RE in human muscle here (Fig. 4B). Evidence in cancer cells indeed suggests that *MYC* transcription is regulated by DNA methylation status (Cheah et al, 1984; de Souza et al, 2013; Kaneko et al, 1985; Rao et al, 1989;

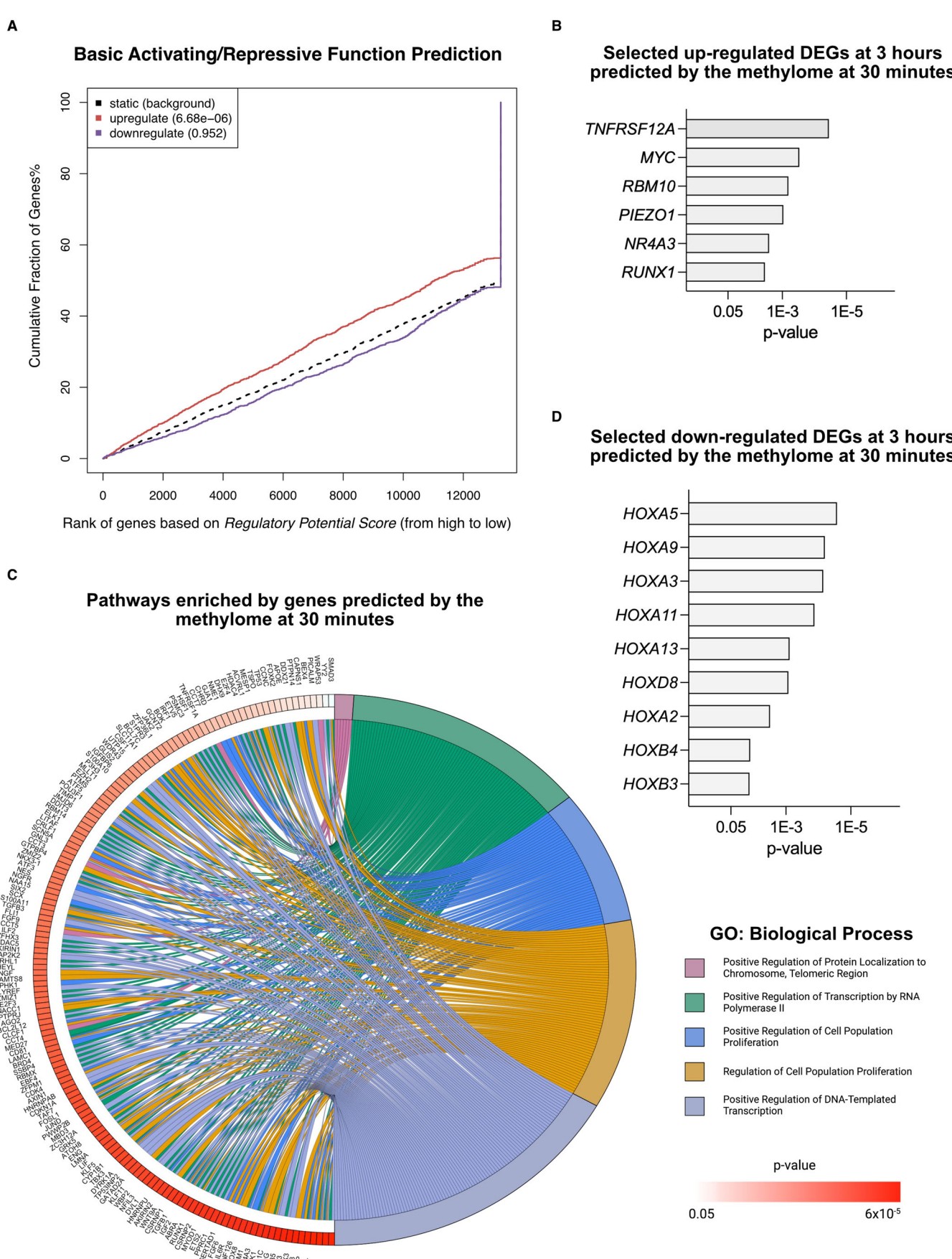

**A** Basic Activating/Repressive Function Prediction

**B** Selected up-regulated DEGs at 3 hours predicted by the methylome at 30 minutes

**C** Pathways enriched by genes predicted by the methylome at 30 minutes

**D** Selected down-regulated DEGs at 3 hours predicted by the methylome at 30 minutes

**Figure 4. Immediate (30 min post-RE) DNA methylome changes after RE predict transcriptional regulation at 3 h.**

(A) Binding and expression target analysis (BETA) combining the DNA methylome at 30 min to differentially expressed genes (DEGs) at 3 h post resistance exercise (RE). BETA integration analysis of up- and downregulated genes to RRBS methylation performed relative to background with predictive interaction significance represented by *p*-values in parenthesis. (B) Selection of upregulated differentially expressed genes (DEGs) at 3 h post-RE significantly predicted by the methylome at 30 min. (C) Chord plot illustrating the top five biological processes (gene ontology) regulated at 3 h post exercise by genes predicted by the methylome at 30 min post exercise. Gene set-associated genes are ordered according to their *p*-values. (D) Selection of downregulated DEGs at 3 h post RE suggested being affected by methylation changes at 30 min post RE. (A–D) was calculated using a one-tailed Kolmogorov–Smirnov test, while Gene ontology (GO) gene set enrichment analysis for (C) was analyzed using a Fisher exact test with Benjamini–Hochberg *p*-value correction. Source data are available online for this figure.

Tsujiuchi et al, 1999), in addition to regulation by other epigenetic layers (Fan et al, 2016; Lüscher, 2001) and G-quadruplexes (Brooks and Hurley, 2010). Of the 936 methylation-controlled upregulated genes predicted by BETA, 155 were coding for five biological processes as suggested by gene set enrichment analysis (Fig. 4C).

BETA integration of 30-min methylome responses with 3-h transcriptome responses to RE was not significant overall for downregulated genes. However, gene-by-gene analysis revealed methylation control for widespread downregulation of HOX genes —*HOXA2*, *HOXA3*, *HOXA5*, *HOXA9*, *HOXA11*, *HOXA13*, *HOXB3*, *HOXB4*, and *HOXD8* (Fig. 4D). In muscle, *HOX* genes are highly regulated by DNA methylation (Tsumagari et al, 2013), and are methylation hotspots during aging that are influenced at the methylation and mRNA levels by physical activity in humans (Turner et al, 2020; Voisin et al, 2021). We previously reported methylation changes around *HOX* genes in myonuclei during hypertrophy (Murach et al, 2021b) and with exercise during aging in muscle tissue (Chambers et al, 2024; Murach et al, 2021a). *HOX* genes control muscle development (Alvares et al, 2003; Poliacikova et al, 2021), but little is known about their role in RE adaptation in adult skeletal muscle.

## MYC governs the late stage RE response via several processes

Due to the overall dominance of upregulated genes coding for ribosomal biogenesis and RNA-binding (Fig. 3A,B), we asked which transcription factors may be steering transcription toward these gene sets. To answer this, we first ran an epigenetic Landscape In Silico deletion Analysis (Lisa) (Qin et al, 2020) on all upregulated genes across the 24-h time course (Dataset EV2). We previously validated the accuracy of this computational approach for *Myc* in skeletal muscle (Jones et al, 2022; Murach et al, 2022). The five top transcription factors influencing the totality of the 24-h recovery period were *NEFLA*, *BCL3*, *FOS*, *MYC*, and *ATF3* (Fig. 5A). Since ribosome-related gene expression primarily dominated late-stage recovery at 8 and 24 h (Fig. 3D,E,G,H), we modeled which transcription factors were controlling expression of DEGs upregulated at the later time points of recovery. The influence of *MYC* on transcription coincided with the transcriptome shift towards the ribosome (Appendix Fig. S1). *MYC* was the number one transcription factor for genes expressed in the later stages of recovery—that is, genes exclusively expressed at 8 and 24 h relative to Pre (Fig. 5B).

Next, we compared the human 24-h post-exercise transcriptional landscape to our previously published datasets on MYC overexpression in muscle of mice (Jones et al, 2022; Murach et al, 2022). Briefly, for these experiments, we generated a doxycycline-inducible muscle-specific model of pulsed MYC induction, called HSA-MYC (human skeletal actin reverse tetracycline transactivator tetracycline response element "tet-on" MYC) (Jones et al, 2022; Murach et al, 2022). Twelve hours of doxycycline in drinking water, followed by 12 h of non-supplemented water, causes upregulation of MYC protein in skeletal muscle (Jones et al, 2022). MYC protein returns to baseline levels after 24 h of drinking un-supplemented water (Appendix Fig. S2). We profiled the transcriptome in the plantaris and soleus muscles 12 h after doxycycline administration (Jones et al, 2022; Murach et al, 2022). Of the 2399 upregulated DEGs induced by RE over 24 h, 316 upregulated genes overlapped the response elicited in the mouse soleus muscle by a single MYC pulse (Fig. 5D–F, Dataset EV4; Jones et al, 2022). Removing the overlapping genes from the human RE response subsequently steered the transcriptional landscape away from the ribosome, as indicated by gene set enrichment analysis (GO: biological processes) on the remaining 2083 DEGs (Fig. 5D). Consequently, using the same gene set enrichment analysis on the 316 genes overlapping the human RE response and soleus transcriptome from the MYC overexpression data generated gene sets largely related to the ribosome (Fig. 5F–H)—specifically, genes coding proteins involved in ribosome biogenesis, assembly, and translation initiation and elongation (e.g., *EEF* and *EIF* genes). The gene expression time course of ribosome biogenesis-related genes under the influence of MYC largely reflected the time point-specific Lisa analysis, suggesting MYC's influence is greatest at 8 h of recovery (Fig. 5G,H). The 672 DEGs exclusive to the MYC induction mouse mainly regulated genes associated with acute changes to transcriptional and translational speed (Fig. 5E). Regulation of overlapping genes was also evident, albeit to a lesser extent, when comparing the human RE response to the smaller MYC-mediated transcriptional response in the plantaris muscle (Fig. EV5; Murach et al, 2022).

Beyond regulation of the ribosome, other genes upregulated by both MYC induction and RE (both with adj. *p* < 0.05) included those involved in actin folding by CCT/TriC (*CCT2*, *CCT3*, *CCT4*, *CCT5*, *CCT6A*, *CCT7*, *CCT8*, *TCP1*), a chaperonin complex that controls sarcomere assembly and organization in striated muscle (Berger et al, 2018; Melkani et al, 2017). Genes associated with metabolism of nucleotides (*AMPD2*, *AK6*, *GART*, *IMPDH1*, *IMPDH2*, *NME1*, *NME2*, *PPAT*, *UCK2*), autophagy (*ATG3*, *HSF1*, *HSPA8*, *HSP90AA1*, *PGAM5*, *TOMM5*, *TOMM22*, *TOMM40*), translation initiation (*EIF1AD*, *EIF2S1*, *EIF2S2*, *EIF3B*, *EIF4A1*, *EIF4A3*, and *EIF5B*), as well as RNA helicases (*DDX21*, *DDX24*, *DDX31*, *DDX54*, *DDX56*, *DHX15*, *DHX30*, and *DHX33*) were also upregulated (Dataset EV4).

Downregulated genes shared by RE in humans and MYC induction in mice included *DNMT3A*, a regulator of DNA methylation in skeletal muscle (Small et al, 2021; Villivalam et al,

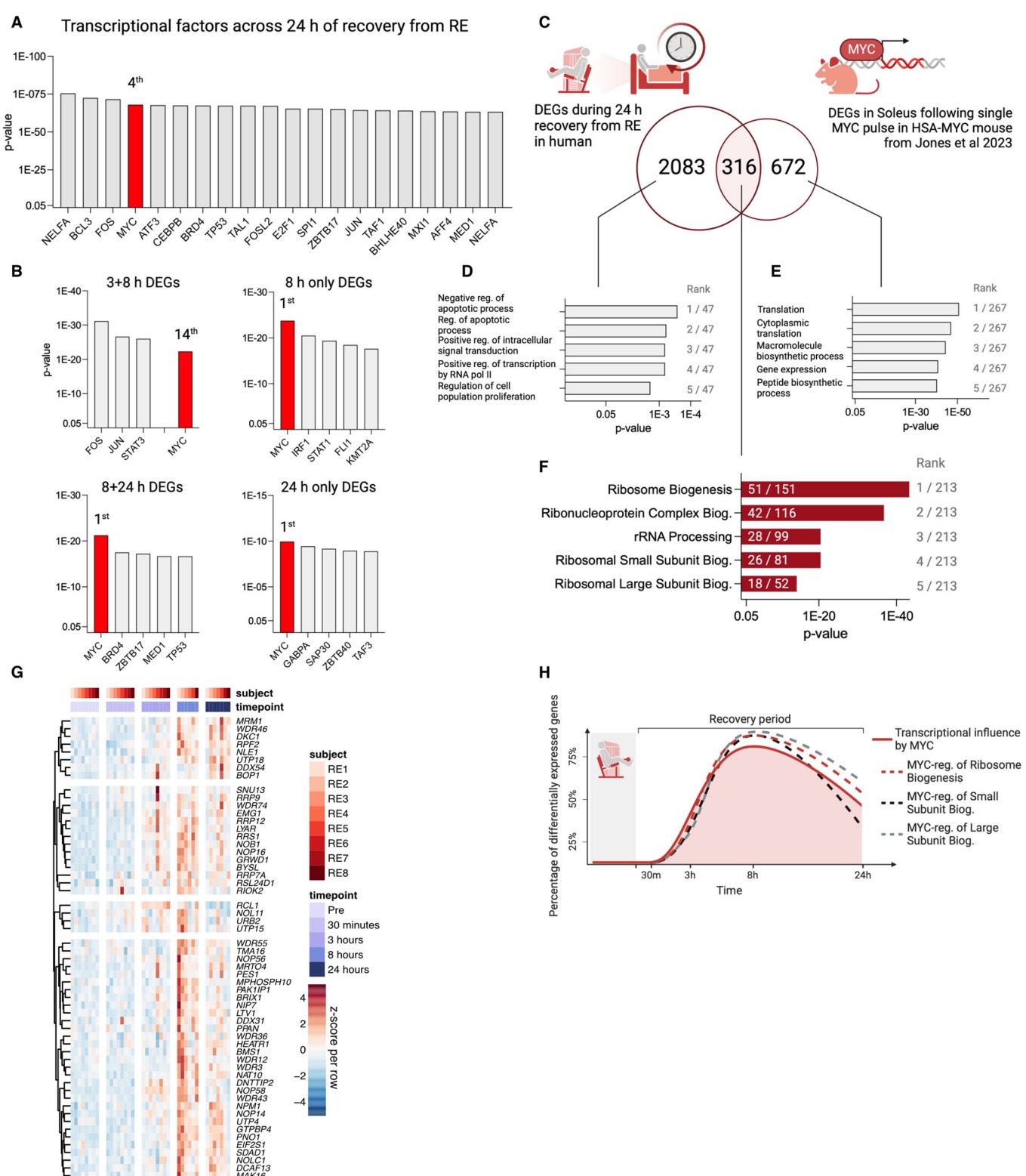

2021), and genes involved in ErbB signaling (*CAMK2G*, *CDKN1B*, *ERBB3*, *GAB1*) (Dataset EV4). Taken together, these data suggest that MYC induction by acute RE in healthy untrained humans may influence the muscle transcriptome in part by directing transcriptional machinery toward the formation of new ribosomes and enhanced translation. MYC may also regulate several other processes involved in skeletal muscle exercise adaptation including actin folding, autophagy, and DNA methylation.

**Figure 5.  The transcription factor MYC dominates late-stage acute recovery from RE by regulating ribosome biogenesis.**

(A) Transcription factors predicted to be active during the 24-h recovery period from resistance exercise (RE) sorted by *p*-value. (B) Transcription factors predicted to regulate the genes expressed exclusively at the later stages of acute recovery. (C) Comparison of upregulated DEGs across 24 h of RE recovery in humans (*n* = 8) vs soleus muscle from MYC-overexpressing mice from Jones et al (2022). (D–F) The top five gene sets (GO: Biological Processes) based on DEGs in (D) the human exclusive gene list, (E) MYC mouse exclusive gene list, and (F) overlapping gene list, respectively. Gene sets are ranked according to their adj. *p*-values. (G) Heatmap showing DEG pattern for ribosome-related genes overlapping human RE response to a MYC response in mouse soleus muscle. Genes retrieved from all five gene sets presented in 3F. (H) Time courses for MYC's transcriptional influence (Red solid line), as well as three MYC-regulated gene sets (dashed lines). (A, B) Is analyzed with epigenetic Landscape In Silico deletion Analysis (Lisa) using a one-sided Wilcoxon rank-sum test. (D–F) Gene ontology (GO) gene set enrichment analysis is analyzed using a Fisher exact test with Benjamini–Hochberg *p*-value correction. Source data are available online for this figure.

## Pulsed muscle fiber specific MYC induction in mice is sufficient for soleus muscle hypertrophy

Our data so far suggests that MYC is a major transcriptional regulator during the acute recovery from RE in human skeletal muscle. An association between skeletal muscle hypertrophy and MYC-controlled acute exercise responses such as enhanced ribosome biogenesis is established (Figueiredo et al, 2021; Hammarström et al, 2020; Stec et al, 2016; von Walden et al, 2012; West et al, 2016), and inhibiting MYC in myotubes blunts ribosome biogenesis and protein synthesis (West et al, 2016). Still, it is unclear whether repeated MYC stimuli alone are sufficient to induce hypertrophy. To address this, we utilized our murine doxycycline-inducible muscle-specific model of pulsatile MYC overexpression: HSA-MYC (Jones et al, 2022; Murach et al, 2022).

We provided doxycycline-supplemented drinking water to 4-month-old female HSA-MYC mice for 48 h, followed by 5 days of un-supplemented water for 4 weeks (five total MYC treatments in *n* = 9 animals). Doxycycline-treated littermate HSA mice were controls (*n* = 7 animals) (Fig. 6A). The doxycycline treatment strategy is similar to the approach from the Belmonte laboratory for overexpressing Yamanaka factors specifically in muscle fibers (Wang et al, 2021). The doxycycline treatment caused MYC to be significantly induced in muscle specifically (Fig. 6B; Appendix Figs. S2 and S3). The 48-h pulse strategy induced a similar amount of MYC protein in the soleus and plantaris muscles and a weaker induction in the gastrocnemius and tibialis anterior muscles; however, the induction was significant across all muscles. The administration pattern was chosen to approximate MYC induction in skeletal muscle by a regular weekly RE regimen.

Pulsed MYC induction resulted in a larger absolute mass (+12.5%, *p* = 0.002; Fig. 6B) and normalized mass (+20.7%, *p* = 0.025; Fig. 6C) of the soleus muscle relative to controls. This magnitude of soleus muscle growth is similar to what is observed after 4 weeks of progressive weighted wheel running (Englund et al, 2020) or 3 weeks of testosterone administration (Englund et al, 2019) in adult female mice. The murine soleus muscle contains a myosin heavy chain (MyHC) fiber type distribution similar to young healthy human vastus lateralis muscle (~50% MyHC I and ~50% MyHC IIa) (Bloemberg and Quadrilatero, 2012; Jones et al, 2022; Murach et al, 2020), which is the muscle from which biopsies were obtained for the current study. The mass of other predominantly fast-twitch mouse hindlimb muscles (containing MyHC 2B and 2X, as well as 2A) was not different with MYC induction versus controls (*p* > 0.05) (Fig. 6E). Likewise, the body weight of the mice was not different between groups (*p* = 0.49, Fig. 6F), nor was food intake in a subset of mice. These data

collectively suggest a muscle and/or fiber-type-dependent effect of MYC for inducing muscle hypertrophy.

To further interrogate this muscle-specific growth, we performed immunohistochemistry on soleus muscle (Fig. 6G). There were no changes in the total amount of fibers within the soleus (Fig. 6H) after pulsatile MYC induction nor were there major shifts in muscle fiber type distribution (Fig. 6I). Overall (+15.1%, *p* = 0.069) and MyHC I fiber cross sectional area (+16.1%, *p* = 0.043) was larger with pulsatile MYC induction relative to controls (Fig. 6J). There was a rightward shift in overall (Fig. 6K) and MyHC I fiber size (Fig. 6L). Fibers expressing MyHC II had a more modest response to pulsatile MYC induction, showing +11.6% difference and a less pronounced rightward shift (*p* = 0.22; Fig. 6J,M). Our prior work showed that the global transcriptional response to a single pulse of MYC is most prominent in the soleus (~1400 DEGs) relative to the plantaris (~500 DEGs) and the quadriceps (<50 DEGs) (Jones et al, 2022; Murach et al, 2022). Given the western blot data presented above across muscle groups, we infer that the soleus muscle is more sensitive to MYC induction than other muscles, specifically the plantaris. These differences in gene expression between muscles likely contributed to soleus-specific mass gains. Given the fiber type and/or muscle-specific effects of *Myc* induction seen in the current and previous work, we revisited our human time course data (Figs. 1 and 2), asking if the degree of *MYC* expression could be related to fiber type distribution. However, no such indications were found, with peak *MYC* expression at 3 and 8 h (Fig. 2C) showing correlations of *r* = 0.37 (*p* = 0.29; Spearman) and *r* = −0.31 (*p* = 0.38; Spearman) vs type I fiber distribution, respectively. Future investigations will probe deeper into MYC dynamics across muscles in our model as well as the specific mechanism(s) by which MYC mediates growth of the soleus. Nevertheless, we provide the first evidence that MYC is sufficient for muscle hypertrophy in the predominant myosin heavy chain fiber types expressed in human skeletal muscle.

## Discussion

The 24-h time course of molecular responses to RE in human skeletal muscle revealed several fundamental aspects of hypertrophic exercise adaptation: (1) the DNA methylome response to RE at 30 min clearly associated with global gene expression at 3 h, (2) a burst of translation and transcription initiation coding transcripts occurs between 3 and 8 h, (3) global gene expression peaks at 8 h after an RE bout, (4) ribosomal-related gene expression dominates the mRNA landscape between 8 and 24 h during

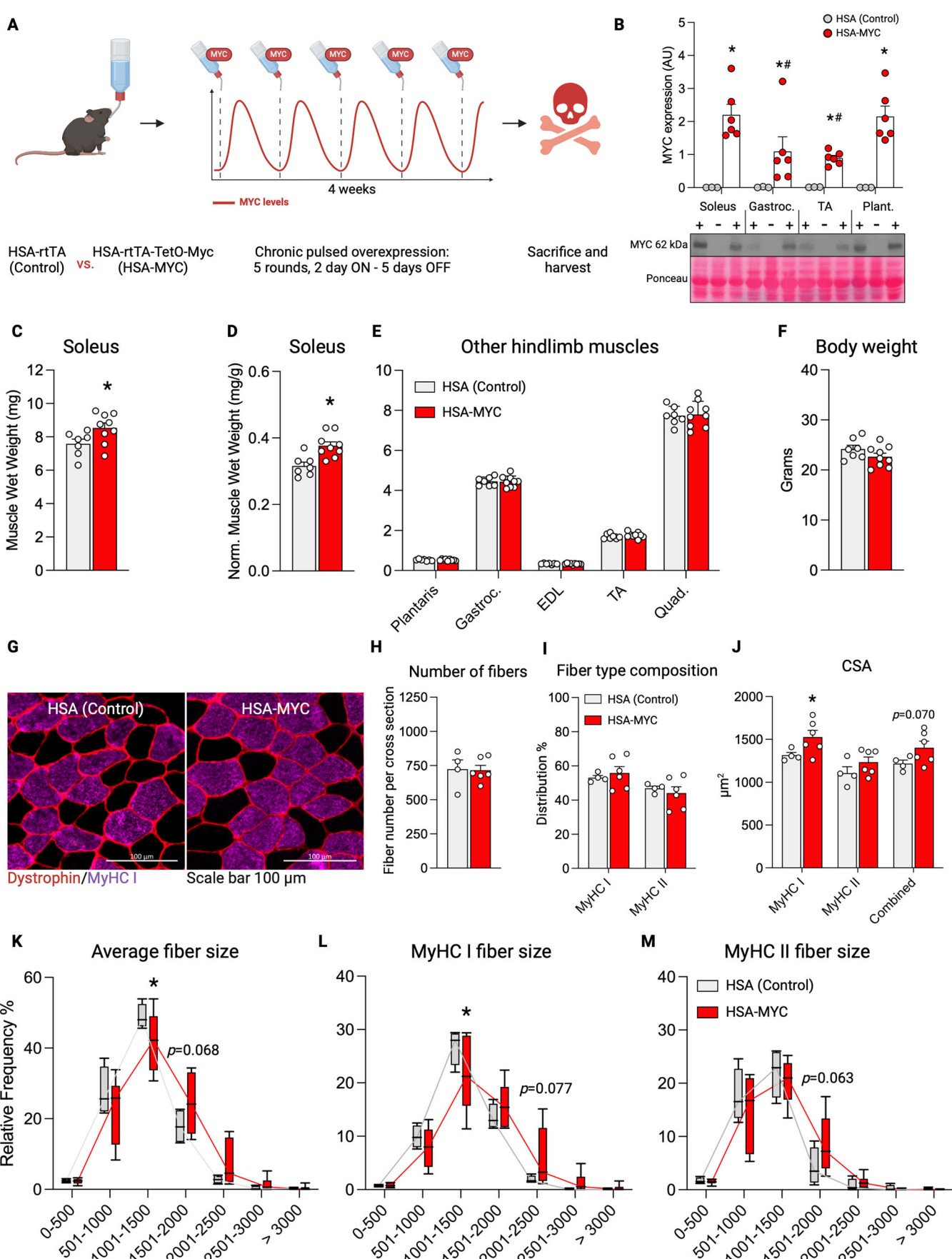

Figure 6.  Four weeks of pulsed MYC induction is sufficient to elicit muscle fiber type specific hypertrophy.

(A) Graphical representation of the experimental design. (B) Soleus ($p = 7.1\text{E}{-}6$ vs Con), Gastrocnemius (Gastroc; $p = 0.0136$ vs Con, $p = 0.0254$ vs Soleus), Tibialis anterior (TA; $p = 0.0422$ vs Con, $p = 0.0048$ vs Soleus), and Plantaris (Plant; $p = 1.0\text{E}{-}5$ vs Con) muscle probed for MYC after 48 h of doxycycline administration. $+$ = HSA-MYC ($n = 6$), $-$ = HSA Control ($n = 3$). (C) Soleus muscle weight of HSA Control and HSA MYC mice after 5 bolus exposures over 4 weeks ($p = 0.0200$ vs Con). (D) Soleus muscle wet weight normalized to body weight ($p = 0.0014$ vs Con). (E) Hindlimb muscle weight of plantaris, Gastroc, extensor digitorum longus (EDL), TA, and quadriceps (Quad.) normalized to body weight. (F) Body weight of mice. (C–F) $n = 7$ HSA Control, $n = 9$ HSA-MYC. (G) Representative images of MyHC I (purple) and MyHC II (black) muscle fiber size in HSA Control (left) and HSA-MYC (right) mice. Dystrophin is outlining fiber borders (red). Scale bar is 100 μm. (H) Total number of fibers per soleus muscle. (I) Distribution of MyHC I and MyHC II fibers expressed as a percentage. (J) Cross sectional area of MyHC I ($p = 0.0433$ vs Con) and MyHC II fibers, and their combined average. (K–M) Frequency distribution plot for average CSA (K; $p = 0.0486$ at 1001–1500 μm range vs Con) and fiber type size of MyHC I (L; $p = 0.0171$ at 1001–1500 μm range vs Con) and MyHC II (M) fibers. (G–M) $n = 4$ HSA Control, $n = 6$ HSA-MYC. Dots are biological replicates. (B–F, H–J) Values represent mean ± SEM. (K–M) The box represents the 25th–75th percentile, the line represents the median, and the whiskers represent Min to Max. *$p < 0.05$ vs control, #$p < 0.05$ vs soleus. Analyzed using Two-way ANOVA with Bonferroni posthoc (B) or Fisher's LSD (K–M), Welch's T-test (C–J). MyHC myosin heavy chain, CSA cross sectional area. Source data are available online for this figure.

recovery after RE, (5) MYC is predicted as a highly influential transcription factor throughout the 24 h recovery period and plays a primary role in ribosomal and translation-related transcription between 8 and 24 h, as well as other processes such as autophagy and actin folding, and (6) periodic pulses of MYC are sufficient to drive muscle growth in the mixed fiber type soleus muscle of female mice.

Two to four hours after RE is typically considered the ideal time to detect changes in gene expression in skeletal muscle (Dickinson et al, 2018; Lavin et al, 2023; Louis et al, 2007; Raue et al, 2012; Yang et al, 2005). By contrast, our data show that the largest number of protein coding DEGs is detected 8 h into recovery. These new findings may influence the design of future RE studies that aim to evaluate gene expression and inform when single time-point biopsies should be taken to interrogate specific recovery processes (e.g., ribosome biogenesis versus MAPK gene expression versus ubiquitin and apoptotic gene expression). Using the same human muscle samples from this study, we previously reported that ribosome biogenesis peaks at 3 h after RE, recovers at 8 h, then rises again at 24 h (Figueiredo et al, 2021). The induction of ribosomal-related mRNAs between 8 and 24 h likely relates to the biphasic increase in rRNA that may support translational capacity for muscle growth (von Walden, 2019; Wen et al, 2016). To this point, our data reveal unique and sometimes multimodal or divergent patterns of gene expression across gene categories over the 24-h time course of recovery after RE. These patterns lend perspective to instances where opposite results in specific gene responses at different post-exercise time points are reported (Amar et al, 2021; Pillon et al, 2020).

Previous studies report mixed findings regarding the agreement between methylation and gene expression in skeletal muscle with exercise training in humans; the relationship may be weak (Robinson et al, 2017) or relatively strong (Lindholm et al, 2014; Seaborne et al, 2018b). With acute exercise, however, the acute methylation response may predict mRNA levels when specifically evaluating the promoter of exercise-responsive genes (Barres et al, 2012; Turner et al, 2019). Using a time course approach with high temporal resolution and a novel and holistic -omic integration technique, we show a strong relationship between the 30-min post-exercise global methylome response and the 3-h post-exercise transcriptome of upregulated genes in skeletal muscle from recreationally active but untrained individuals. This relationship may change with additional structured training, however. For instance, the *Myc* response to repeated bouts of RE tends to become

blunted over time (Viggars et al, 2022a); this may contribute to hypertrophic response heterogeneity between individuals (Lavin et al, 2021; Phillips et al, 2013; Sparks, 2017; Stec et al, 2016). BETA analysis predicted MYC transcription to be regulated at the epigenetic level, consistent with work in cancer cells (Cheah et al, 1984; de Souza et al, 2013; Kaneko et al, 1985; Rao et al, 1989; Tsujiuchi et al, 1999). Our current and previous findings suggest that *Myc* is regulated by DNA methylation status in muscle fibers during hypertrophy (von Walden et al, 2020). More work is needed to determine whether epigenetic changes underpin lower transcriptional sensitivity of *Myc* with repeated bouts (Viggars et al, 2022a), as well as reduced training responsiveness between individuals (Stec et al, 2016). Another possibility is that the timing of transcriptional responses to acute exercise in the trained state experiences a "phase shift" relative to untrained muscle, and that this is related to "priming" by altered DNA methylation. Such a phenomenon was recently described in mouse muscle after endurance exercise training (Furrer et al, 2023). There is a need for time course studies in humans evaluating the molecular responses to RE in the trained versus untrained state so that correct interpretations can be made regarding differential expression of genes (such as *MYC*) versus differential timing of expression. Regardless, our current findings reinforce the evidence for acute exercise responses operating under the control of early methylation events (Barres et al, 2012; Seaborne et al, 2018a) and support data suggesting that methylation changes are central to exercise training adaptations in humans (Egan and Sharples, 2022; Roberts et al, 2023; Sharples and Seaborne, 2019; Turner et al, 2019).

Our work points to MYC as a key player in controlling hypertrophic adaptation to exercise in skeletal muscle. In mice, *Myc* is actively transcribed and enriched in myonuclei during mechanical overload (Murach et al, 2022). A single pulse of *Myc* in skeletal muscle markedly alters the transcriptome (soleus>plantaris>quadriceps) (Jones et al, 2022; Murach et al, 2022) and rewires the DNA methylation landscape (Jones et al, 2022). In humans, MYC may act directly on rDNA after RE to influence ribosome biogenesis (Figueiredo et al, 2021), consistent with MYC's occupation of the rDNA promoter during load-induced muscle hypertrophy (von Walden et al, 2012). MYC controls ribosome biogenesis as well as skeletal muscle protein synthesis independent from mTORC1 activation (Mori et al, 2020; West et al, 2016), but its ability to drive muscle growth has been unclear (Mori et al, 2020; Phillips et al, 2013). It is likely that chronic induction of MYC in muscle is detrimental to mass and function, similar to what

occurs with prolonged chronic mTORC1 activation (Castets et al, 2013; Ham et al, 2020). By using a doxycycline-inducible and pulsatile model of MYC induction in skeletal muscle, we show for the first time that MYC can promote growth of skeletal muscle mass in the murine soleus. Our prior and current work suggests this hypertrophy could be due to ribosomal regulation—biogenesis, efficiency, and/or specialization (Jones et al, 2022; Murach et al, 2022). Larger muscle size caused by MYC may also be attributable to the regulation of several other processes such as actin folding and/or autophagy among others. Transcriptome data from MYC induction in the soleus indicates that a variety of other processes contribute to hypertrophy since >1300 genes are altered by a single MYC pulse (Jones et al, 2022). Our findings encourage further investigation of how pulsatile MYC supports long-term anabolism at the molecular, signaling, and cellular level across muscles, fiber types, sexes, and ages.

Collectively, the time course of -omics responses to RE in healthy untrained humans, alongside the repeated biopsy control group, is a valuable resource to the skeletal muscle field. Our results define the molecular landscape after exercise at high temporal resolution and will help inform the design of future human exercise studies. We detail the interplay between the methylome and transcriptome, identify MYC as a key component of the RE response, and show that MYC is sufficient for muscle growth. This study complements previous and ongoing efforts at defining the acute muscle -omics responses to RE in humans (Amar et al, 2021; Pillon et al, 2020; Sanford et al, 2020) to uncover new molecular regulators of hypertrophic physical activity. We lay the groundwork for future investigations that will expand on how transcriptional regulators such as *MYC* control muscle mass and adaptation in skeletal muscle.

# Methods

**Reagents and tools table**

| Reagent/Resource | Reference or Source | Identifier or Catalog Number |
|---|---|---|
| **Experimental Models** | | |
| Human skeletal actin reverse tetracycline transactivator tetracycline response element "tet-on" HSA-MYC Mice | Jackson Laboratory | Strains 038301 & 019736 |
| **Recombinant DNA** | | |
| **Antibodies** | | |
| Dystrophin primary antibody | Abcam, St. Louis, MO, USA | ab15277 |
| MyHC 1 | Developmental Studies Hybridoma Bank, Iowa City, IA, USA | BA-D5 |
| Anti-MYC | Cell Signaling, Danvers, MA, USA | D84C12 cat. 5605 |
| Secondary antibody utilized for MYC stain | LI-COR Biosciences, Lincoln, NE | IRDye 800CW/ 680RD |
| **Oligonucleotides and other sequence-based reagents** | | |
| **Chemicals, Enzymes and other reagents** | | |

| Reagent/Resource | Reference or Source | Identifier or Catalog Number |
|---|---|---|
| TRI Reagent | Sigma-Aldrich, St Louis, MO, USA | |
| Doxycycline hyclate | Sigma | D9891-5G |
| **Software** | | |
| MyoVision | Viggars et al, 2022b; Wen et al, 2017 | |
| HISAT2 (2.0.5) | Kim et al, 2015 | |
| featureCounts (1.5.0-p3) | Liao et al, 2014 | |
| R (Version: 4.1.0) | https://cran.r-project.org/bin/windows/base/old/4.1.0/ | |
| DESeq2 (1.42.0) | Love et al, 2014 | |
| org.Hs.eg.db: Genome wide annotation for Human. R package (version 3.8.2) | Carlson et al, 2019 | |
| CIBERSORTx | Newman et al, 2019 | |
| Seurat analysis pipeline | Hao et al, 2021 | |
| methylKit R package | Akalin et al, 2012 | |
| Enrichr with 2023 gene ontology database | https://maayanlab.cloud/Enrichr/ 2023-Aug-16 Aleksander et al, 2023; Chen et al, 2013; Kuleshov et al, 2016; Xie et al, 2021 | |
| Landscape In Silico deletion (LISA) Analysis | Qin et al, 2020 | |
| Venny (2.1) | https://bioinfogp.cnb.csic.es/tools/venny/index.html 2023-Aug-23 | |
| BETA Analysis | Wang et al, 2013 | |
| GraphPad Prism (version 7.00 for Mac OS X) | GraphPad Software, La Jolla, CA | |
| Rstudio | https://posit.co/download/rstudio-desktop/ | |
| BioRender | https://www.biorender.com/ | |
| Affinity Designer (2.3) | https://affinity.serif.com/en-us/designer/ | |
| **Other** | | |
| Illumina NovaSeq 6000 | 150 bp paired-end sequencing; Novogene Corp. Inc., Sacramento, CA, USA | |
| Single-cell RNA sequencing data from Lovrić et al (GSE214544) | Lovrić et al, 2022 | |
| Previously published reduced representation bisulfite sequencing (RRBS) for ribosomal DNA (rDNA) | Figueiredo et al, 2021 | |
| Trim Galore Wrapper (FastQC & Cutadapt) | https://github.com/FelixKrueger/TrimGalore | |
| Published gene- and fiber-type area distribution data | Reitzner et al, 2024 | |
| RC/DC Protein Assay | BioRad, Hercules, CA, USA | cat. 500-0119 |

## Ethical approval

The regional Ethical Review Board in Linköping (2017/183-31) approved the study protocol for the human intervention. The volunteers received oral and written information about the study, and subsequently provided their informed consent prior to study enrollment. The study protocol conformed with the Declaration of Helsinki. IACUCs at the University of Arkansas (UA, AUP 21038) approved all animal procedures. Mice were housed in a temperature and humidity-controlled room, maintained on a 12:12 h light:dark cycles, and food and water were provided ad libitum throughout experimentation. All animals were sacrificed via cervical dislocation under deep anesthesia with inhaled isoflurane.

## Volunteers

A subset of thirteen volunteers was chosen for analysis from previously published studies (Figueiredo et al, 2021; von Walden et al, 2021). The subset was chosen based on which participants had the most complete set of biopsy materials still available. Eight recreationally active volunteers were analyzed from the RE group (5m/3f), and five from the CON group (3m/2f). The volunteers in the RE group had a mean age of $32 \pm 5$ years, height of $181 \pm 9$ cm, weight of $83 \pm 8$ kg, and body mass index (BMI) of $25.3 \pm 2.0$. The corresponding values for the CON group were an age of $30 \pm 4$ years, height of $177 \pm 5$ cm, weight of $85 \pm 12$ kg, and a BMI of $27.3 \pm 3.6$.

## Experimental protocol

The experimental protocol has been described elsewhere (Figueiredo et al, 2021; von Walden et al, 2021). In short, volunteers were instructed to not partake in any strenuous physical activity for the legs for 3 days prior to the intervention. Following an overnight fast, subjects consumed a breakfast consisting of a standardized amount of liquid formula supplying 1.05/0.28/0.25 grams of carbohydrates/protein/fat per kg of body weight (Resource Komplett Näring, Nestlé Health Science, Stockholm, Sweden). Skeletal muscle biopsies were collected from the vastus lateralis, using a Bergström needle with manually applied suction (Evans et al, 1982). Ninety minutes after breakfast, volunteers started a 45-min standardized RE session. The RE session consisted of a short warm-up on a cycle ergometer, followed by four sets at 7RM load with two min of rest using both leg press and leg extension machines. Muscle biopsies were collected 1 h after breakfast (Pre), as well as 30 min and 3 h after RE completion. Between the completion of the exercise and the 3-h biopsy, volunteers were resting in a seated position under the supervision of the test leaders. Immediately following the 3-h biopsy, another portion of the standardized liquid formula was administered for lunch to the volunteers (2.1/0.56/0.5 grams of carbohydrates/protein/fat per kg of body weight). Following the standardized lunch, volunteers were allowed to go home but were instructed to refrain from physical activity and food intake. Eight hours after RE ended, volunteers reported to the laboratory again, and another muscle biopsy was collected whereby the volunteers were sent home overnight. At home, volunteers were instructed to eat a standard dinner (a balanced meal of ~25% of a protein source and equal distribution of carbohydrate sources and greens) in the evening. Volunteers were

again instructed to refrain from any physical activity other than light work. Volunteers got clear instructions to follow but were not monitored during the rest period between the 3-h and 8-h, as well as the 8-h and 24-h sampling. Another administered standardized liquid formula breakfast was ingested the following morning 90 min prior to reporting to the laboratory for the final muscle biopsy sampling 24 h after RE completion (breakfast ingested 2 h prior to biopsy sampling). The experimental protocol is depicted in Fig. 1A (CON) and Fig. 2A (RE). The sampling time points for the CON group was matched to the exercise group.

## RNA extraction, sequencing, and analysis

Approximately 25 mg of muscle tissue was homogenized in TRI Reagent (Sigma-Aldrich, St Louis, MO, USA) using a Bullet Blender Tissue Homogenizer (Next Advance, Troy, NY, USA). An RNA supernatant phase was then isolated using bromochloropropane and centrifugation. Next, the RNA phase was processed using Direct-zol filter columns (Zymo Research, Irvine, CA, USA). Finally, the RNA was treated with DNAse and eluted in DEPC-treated water prior to storage at $-80\,°C$. Concentration, and purity of the RNA was determined using a BioTek PowerWave XS microplate reader (BioTek Instruments Inc., Winooski, VT, USA). Library preparation of mRNA was done using Poly A enrichment, followed by RNA sequencing by an Illumina NovaSeq 6000 (150 bp paired-end sequencing; Novogene Corp. Inc., Sacramento, CA, USA).

Quality control of raw sequencing reads was performed by removing adapters and low-quality reads. Subsequently, the reads were aligned to the human reference genome (GRCh38.p12) using HISAT2 (2.0.5) (Kim et al, 2015), and the quantification of reads mapped to each gene was conducted using featureCounts (1.5.0-p3) (Liao et al, 2014). Raw counts were used as inputs for the downstream analysis in R platform (Version: 4.1.0). After filtering out genes with low expression, DESeq2 (1.42.0) was used for the normalization and differential analyses in the comparison between different time point groups (Love et al, 2014). Genes with a false discovery rate (Benjamini–Hochberg method) adjusted $p$-value < 0.05 were identified as differentially expressed genes (DEGs). We have used org.Hs.eg.db (3.13.0) as reference for the annotation of human genes (org.Hs.eg.db: Genome wide annotation for Human. R package version 3.8.2.) (Carlson et al, 2019). To determine gene expression patterns among the DEGs, we computed z-score per gene along different time points for each group separately and employed the Euclidean hierarchical clustering method to identify clustered genes. The total number of clusters was determined empirically. Raw and processed files have been deposited in the GEO database (GSE252357).

## Data deconvolution

Leveraging single-cell RNA sequencing data, the relative abundance of different cell types from bulk tissue transcriptomes was estimated using the computational tool CIBERSORTx (Newman et al, 2019). Single-cell RNA sequencing data from Lovrić et al (GSE214544) (Lovrić et al, 2022) were used for constructing the reference matrix of the analyses. The original datasets based on 10X Genomics technology, were reanalyzed using the Seurat pipeline (Hao et al, 2021). Different cell types were separated with the resolution parameter set to 0.5 and then annotated based on marker

genes from the previous publication (Hao et al, 2021). For determination of mononuclear cells, cell populations annotated as "myocytes" were excluded from the analyses as they dominate the transcriptome in the deconvolution. Subsequently, the normalized gene expression data of individual cells were compiled to create a comprehensive signature matrix encompassing the entire spectrum of different cell types. This matrix served as the basis of cell type proportion prediction in CIBERSORTx, with 1000 permutations to ensure the robustness and accuracy of the predictions.

## DNA methylation data processing and statistical analysis

We previously published reduced representation bisulfite sequencing (RRBS) for ribosomal DNA (rDNA) (Figueiredo et al, 2021). Here, we used this RRBS dataset for global DNA methylation analysis (von Walden et al, 2020). Quality control and adapter sequence trimming were performed using FastQC and Cutadapt, respectively as parts of the Trim Galore wrapper. Low-quality base calls (Phred score <20) were removed prior to trimming adapter sequences. Bismark aligner was used to align the sequence reads to the bisulfite-converted GRCh38 genome prior to data processing. Coverage (.cov) files produced from Bismark aligner were used for data analysis in the methylKit R package (Akalin et al, 2012). MethylKit was used to pool samples into their respective groups to maximize read coverage across the genome using a minimum read cutoff of 10 reads per base, and minimum base coverage of 1 sample per group. Differentially methylated regions (DMRs) were determined by genomic ranges for every gene promoter as defined by the hg38.bed file obtained from NCBI. Fisher's exact test with sliding linear model (SLIM) correction for false discovery (Wang et al, 2011) was used to qualify both differentially methylated sites and differentially methylated promoters within the dataset. Percent methylation and percent differential methylation were then obtained from methylKit following analysis.

## Pathway analysis, transcriptional regulators, and BETA analysis

The up- and downregulated DEG (adj. $p < 0.05$) were initially separated. All DEGs from 30-min, 3-, 8-, and 24-h post-exercise biopsies were collapsed into one DEG list across the 24-h recovery period for up- and downregulated genes, respectively. Gene set enrichment analyses were conducted on the collapsed gene lists using Enrichr (https://maayanlab.cloud/Enrichr/ 2023-Aug-16) with the 2023 gene ontology (GO) database as our cross reference (Aleksander et al, 2023; Chen et al, 2013; Kuleshov et al, 2016; Xie et al, 2021). We used all protein-coding genes detected in our muscle samples (14,392 genes, Figs. EV1 and 2) as our background correction for the pathway analysis, as suggested by Stokes et al (Stokes et al, 2023). The output for gene sets within the Biological Process and Molecular Function of the Gene Ontology database are reported as source data, and the number of genes, and adjusted $p$-values of selected enriched gene sets are presented in Fig. 2. The time course analysis for each gene set is based on the proportion of DEGs within each gene set in the current dataset expressed at each biopsy time point relative to Pre. Thus, the number of DEGs within a specific gene set per time point is divided by the number of DEGs within the same gene set from the pooled 24-h DEG list, described above.

Landscape In Silico deletion Analysis (Lisa) was run according to the recommended procedures as reported by Qin et al (Qin et al, 2020), consistent with our previous work (Jones et al, 2022; Murach et al, 2022). In brief, DEG lists (adj. $p < 0.05$) were run using software on http://lisa.cistrome.org (2023-Aug-20). If the number of DEGs was above 500, the analysis was run locally using the command line. Lisa analysis was performed on the collapsed upregulated 24-h gene list, and on upregulated genes expressed at different time points. The Cauchy combination $p$-value test was used to determine the overall influence of MYC. Overlapping DEGs between the human RE response and the MYC mouse was analyzed using Venny2.1 (https://bioinfogp.cnb.csic.es/tools/venny/index.html 2023-Aug-23).

We presented the method for incorporation of RRBS and RNA-seq data using BETA in a previous publication (Ismaeel et al, 2023). BETA is a software that provides an integrated analysis of transcription factor binding to genomic DNA and transcript abundance using chromatin immunoprecipitation sequencing (ChIP-seq) and transcriptomics (RNA-seq) datasets (Wang et al, 2013). BETA takes into consideration the distance of the regulatory element relative to the transcription start site (TSS) by modeling the effect of regulation using a natural log function termed the regulatory potential (Eq. 1), as described previously by Tang and colleagues (Tang et al, 2011). CpG islands were converted into "methylation peaks" similar to transcription factor binding peaks, which is built using the GRCm39 CpG island bedfile downloaded from the UCSC genome browser. Only genes differentially expressed with adjusted $p < 0.05$ from RNA sequencing analysis were included as input for gene expression. The BETA basic command was run with the following parameters "-c 0.05 --df 0.05 --da 500".

$$s_g = \sum_{i=1}^{k} e^{-(0.5+4\Delta_i)} \tag{1}$$

In the current study, a CpG island ($k$) within 100 kb of TSS of gene ($g$) is included in the calculation for the regulatory potential score ($s$). The distance between the CpG island and the TSS is expressed as a ratio relative to 100 kb ($\Delta$). The scoring is weighted based on the distance from the TSS (higher for smaller distances, lower for larger distances).

## BETA integration gene set enrichment analysis

Gene set analysis was performed using the enrichR R package. Up- and down-target genes resulting from BETA integration of DNA methylation and RNA-sequencing datasets were included in independent up and down gene sets. The 2023 Gene Ontology database Biological Processes was used to annotate up and down gene sets and determine enriched ontologies for each gene set. GOplot R package was used to combine log2 fold change for each gene with their respective enriched gene sets.

## Interpolation of fiber type distribution in muscle samples

Published gene- and fiber-type area distribution data from Reitzner et al (Reitzner et al, 2024) were obtained. A ratio of *MYH7* to *MYH1* & *MYH2* normalized counts was created for each baseline sample ($MYH7/(MYH1 + MYH2)$). Next, a standard curve was created for estimating fiber type distributions based on the gene ratio (Fig. EV4A). An identical gene ratio ($MYH7/(MYH1 +$

MYH2)) was created for the baseline samples in our dataset, and type I fiber area % was subsequently interpolated using the standard curve created from the values obtained from Reitzner et al (Reitzner et al, 2024).

## Generation of HSA-MYC mice and in vivo pulsatile overexpression experiments

Human skeletal actin reverse tetracycline transactivator tetracycline response element "tet-on" MYC (HSA-MYC) mice were generated as previously described (Jones et al, 2022; Murach et al, 2022) (Jackson Laboratory Strains 038301 and 019736). A subset of mice ($n = 8$) was crossed with tet-on green fluorescent protein mice for myonuclear isolation experiments not presented here (Jackson Laboratory Strain 005104). For all MYC experiments, littermate mice (HSA or HSA-GFP) were controls; all mice were heterozygous for each transgene. At four months of age, control ($n = 7$) and MYC overexpressing female mice ($n = 9$) were given doxycycline in drinking water with sucrose (0.5 mg mL$^{-1}$ with 2% sucrose) for 48 h. All mice were then given un-supplemented drinking water for the remaining 5 days of the week. This dosing strategy was repeated 5 total times. All mice were euthanized 24 h following the final doxycycline treatment. Some mice were used for analyses not described here, so the histology results are from $n = 4$ control and $n = 6$ MYC induction mice. Mice were euthanized in the morning (before 10:00 AM) and all tissues were harvested, weighed, and frozen in liquid nitrogen-cooled isopentane using optimal cutting temperature compound. The average mass of both muscles for every mouse is presented.

## Immunohistochemistry

Fiber cross sectional area and fiber type analyses on the soleus muscles were performed as previously described (Dungan et al, 2022; Murach et al, 2020). Briefly, 8 µm sections were cut using a cryostat and air dried for ≥1 h. Primary antibodies for dystrophin (1:100, ab15277, Abcam, St. Louis, MO, USA) and MyHC 1 (BA-D5, Developmental Studies Hybridoma Bank, Iowa City, IA, USA) were applied for ≥4 h in a PBS cocktail. After several PBS washes, isotype-specific secondary antibodies were applied for 1 h. Following several PBS washes, the slides were mounted with cover slips using a 50/50 solution of PBS and glycerol. Muscle cross sections were imaged using a Zeiss AxioImager M2. Fiber cross sectional area, fiber number, and fiber type distribution was analyzed using MyoVision (Viggars et al, 2022b; Wen et al, 2017), as we have previously described, using the entire muscle cross-section.

## Western blotting for MYC

Western blots were for MYC were carried out as previously described (Jones et al, 2022). Briefly, 20 mg of frozen muscles and liver were powdered and homogenized in Laemmli buffer. Following RC/DC assay (BioRad, Hercules, CA, USA, cat. 500-0119), 40 µg of total protein was subjected to SDS-PAGE using a 10% gel. Membranes were blocked with 5% of milk. Primary antibody incubation was conducted at 4 °C for ~72 hours using anti-MYC (D84C12 cat. 5605, Cell Signaling, Danvers, MA, USA) diluted 1:500. Secondary antibody (IRDye 800CW/680RD, LI-COR Biosciences, Lincoln, NE) was diluted 1:10,000 and membranes

were imaged on LI-COR Odyssey FC using IR detection. All bands were normalized to the 45-kDa actin band of Ponceau S stain as a loading control.

## Statistical considerations

Figures were generated using GraphPad Prism version 7.00 for Mac OS X (GraphPad Software, La Jolla, CA), Rstudio, BioRender, and Affinity Designer 2.3. Data presented as mean ± standard deviation of mean unless otherwise stated. For the human study, sample sizes were based on tissue availability from a prior investigation where the most complete sample sets were used (Figueiredo et al, 2021). For the murine studies, sample size was based on mouse availability, but the magnitude of increase in soleus mass (~15%) that we previously observed with 4 weeks of murine exercise ($n = 8$/group) (Englund et al, 2020) or 3 weeks of testosterone administration ($n = 6$–8) (Englund et al, 2019) was also used to inform our analysis. Average muscle weights for each animal (mean of both muscles) and histology data were analyzed using two-tailed dependent t-tests with $p < 0.05$. Prior to t-tests, data were analyzed for normality using Shapiro–Wilks formula. For all -omics analyses, Benjamini–Hochberg adjusted $p$ values (adj. $p < 0.05$) were utilized.

## Data availability

RNA-sequencing data were deposited in GEO under accession GSE252357; processed data are provided as source data. A publicly available user-friendly web-based application is provided at http://data.myoanalytics.com for browsing the human transcriptional time course data. RRBS was deposited in association with our previous publication (Figueiredo et al, 2021). RRBS data are in GEO: GSE252357.

The source data of this paper are collected in the following database record: biostudies:S-SCDT-10_1038-S44319-024-00299-z.

## Peer review information

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

## Acknowledgements

Thank you to the volunteers for their willingness to participate in this research and contribute their tissue and time. Thank you to Kate Mamiseishvili, PhD, Dean of the College of Education at the University of Arkansas for resources that enabled the rapid completion of the histology for this project. Thank you to Charlotte Peterson, PhD, and John McCarthy, PhD, of the University of Kentucky for their support of this work. Thank you to Andrew P. McMahon, PhD, of the University of Southern California who provided the "tet-o" *Myc* mice as a generous gift. Thank you to Stefan Reitzner, PhD, Eric Emanulesson, and Carl-Johan Sundberg, PhD, of the Karolinska Institute for kindly providing data to enable fiber type interpolation. This work was supported by National Institutes of Health R00 AG063994, R01 AG080047, and funds from the University of Arkansas Vice Chancellor for Research and Innovation as well as the College of Education and Health Professions to KAM. This research was conducted while KAM was a Glenn Foundation for Medical Research and AFAR Grant for Junior Faculty awardee. This work was also supported by the Arkansas Integrated Metabolic Research Center – AIMRC (P20GM139768). YW was supported by funding from the National Institute of Health K99 AR081367. SE was supported by Centrum för Idrottsforskning (P2024-0166). FVW was supported by Vetenskapsrådet (2022-01392), AFM-Telethon (23137), Åke WibergStiftelse, Sveriges Läkarförbund, and Centrum för Idrottsforskning (P2023-0137, P2024-0102).

## Author contributions

**Sebastian Edman**: Data curation; Formal analysis; Funding acquisition; Investigation; Visualization; Methodology; Writing—original draft; Project administration; Writing—review and editing. **Ronald G Jones III**: Data curation; Formal analysis; Validation; Visualization; Methodology; Writing—review and editing. **Paulo R Jannig**: Data curation; Software; Formal analysis; Visualization; Writing—review and editing. **Rodrigo Fernandez-Gonzalo**: Resources; Investigation; Writing—review and editing. **Jessica Norrbom**: Resources; Investigation; Writing—review and editing. **Nicholas T Thomas**: Data curation; Software; Formal analysis; Writing—review and editing. **Sabin Khadgi**: Investigation; Writing—review and editing. **Pieter J Koopmans**: Formal analysis; Writing—review and editing. **Francielly Morena**: Formal analysis; Writing—review and editing. **Toby L Chambers**: Validation. **Calvin S Peterson**: Formal analysis; Writing—review and editing. **Logan N Scott**: Formal analysis; Writing—review and editing. **Nicholas P Greene**: Resources; Supervision; Writing—review and editing. **Vandre C Figueiredo**: Methodology; Writing—review and editing. **Christopher S Fry**: Resources; Supervision; Writing—review and editing. **Liu Zhengye**: Data curation; Software; Formal analysis; Validation; Visualization. **Johanna T Lanner**: Resources; Writing—review and editing. **Yuan Wen**: Resources; Data curation; Software; Formal analysis; Supervision; Funding acquisition; Visualization; Methodology; Writing—review and editing. **Björn Alkner**: Resources; Supervision; Investigation; Project administration; Writing—review and editing. **Kevin A Murach**: Conceptualization; Resources; Data curation; Formal analysis; Supervision; Funding acquisition; Validation; Investigation; Methodology; Writing—original draft; Project administration; Writing—review and editing. **Ferdinand von Walden**: Conceptualization; Resources; Supervision; Funding acquisition; Investigation; Methodology; Project administration; Writing—review and editing.

Source data underlying figure panels in this paper may have individual authorship assigned. Where available, figure panel/source data authorship is listed in the following database record: biostudies:S-SCDT-10_1038-S44319-024-00299-z.

## Disclosure and competing interests statement

YW is the founder of MyoAnalytics LLC. The authors have no other conflicts to declare.

# Expanded View Figures

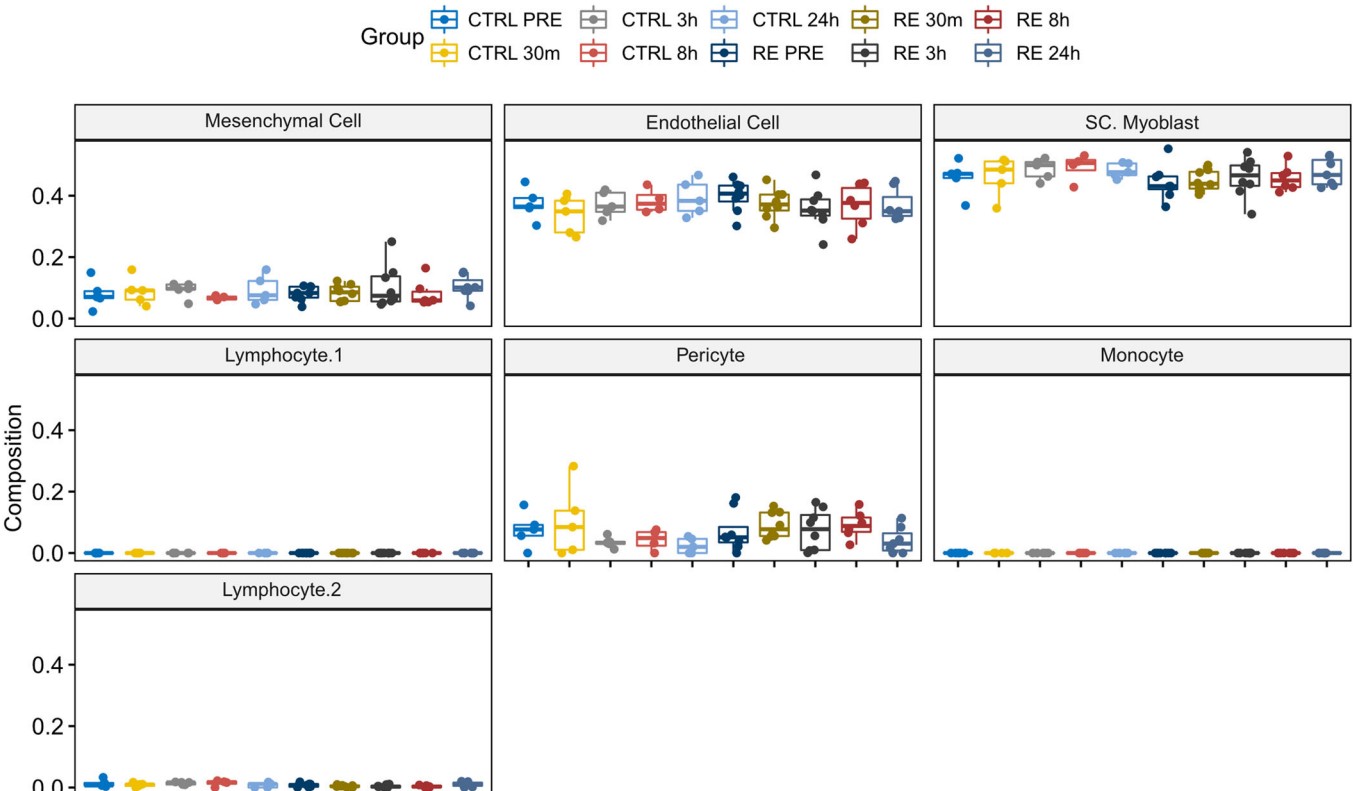

Figure EV1.  **Cell composition of muscle biopsies through data deconvolution.**

Cell composition of all skeletal muscle biopsy time points excluding "myocytes" (CTRL $n = 5$, RE $n = 8$). The box represents the 25th–75th percentile, the line represents the median, and the whiskers represent Min to Max, excluding outliers. SC satellite cell, CTRL control group, RE resistance exercise group.

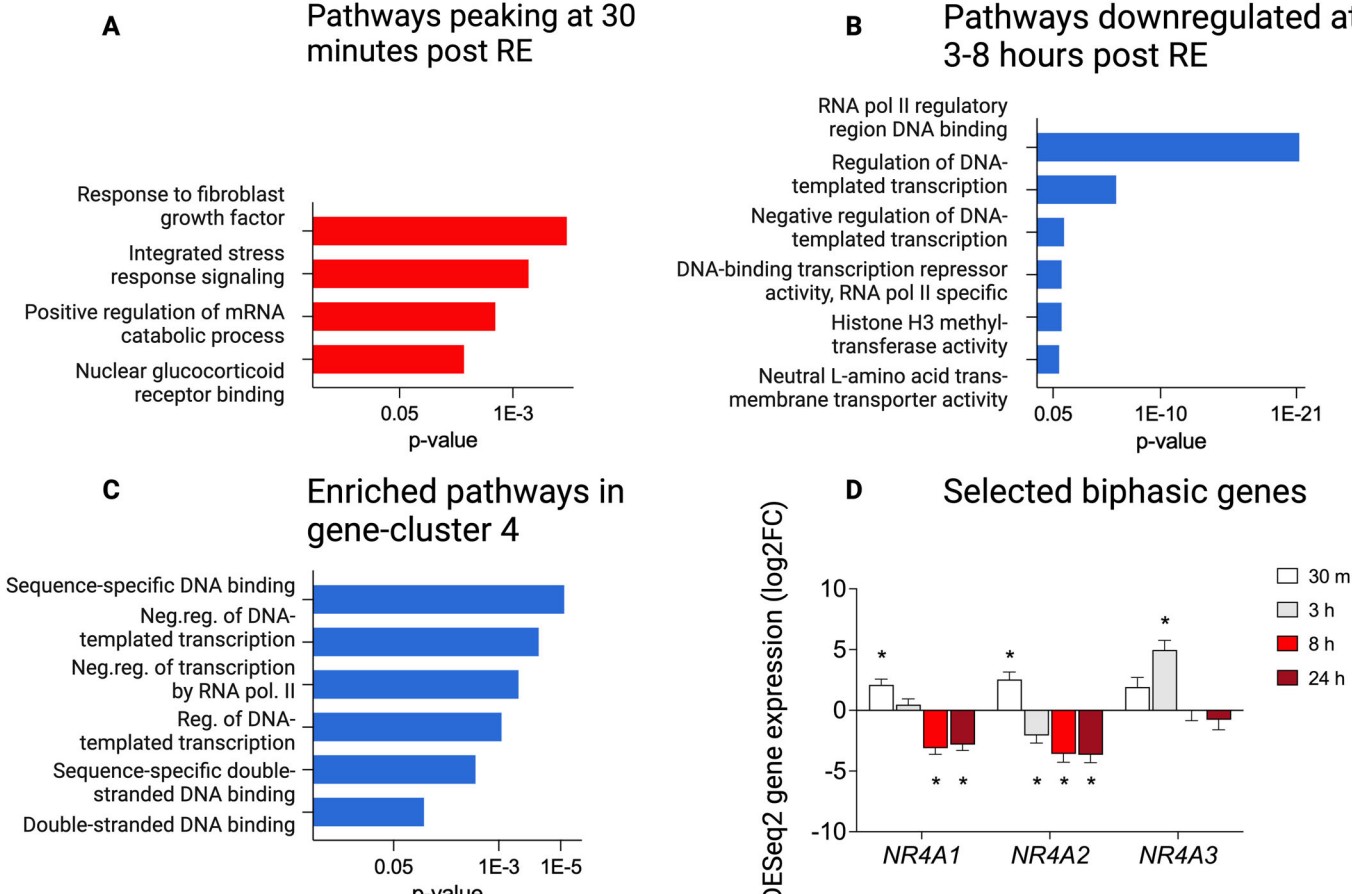

**Figure EV2. Targeted gene set enrichment analysis.**

Pooled gene ontology (GO) biological processes and molecular function gene sets. (A) Targeted analysis of gene sets peaking at 30 min post resistance exercise (RE). (B) Targeted analysis of gene sets significantly enriched in genes downregulated 3–8 h post RE. (C) Targeted analysis of biphasic DEGs composing cluster 4, as presented in Fig. 2D. NR4A1 adj. $p = 0.0049$ at 30 min, adj. $p = 6.5E{-}8$ at 8 h, adj. $p = 1.0E{-}6$ at 24 h, NR4A2 adj. $p = 0.0135$ at 30 min, adj. $p = 0.0081$ at 3 h, adj. $p = 4.7E{-}6$ at 8 h, adj. $p = 2.4E{-}6$ at 24 h, NR4A3 adj. $p = 1.8E{-}8$ at 8 h. (A–C) Gene ontology (GO) gene set enrichment analysis is analyzed using a Fisher exact test with Benjamini–Hochberg $p$-value correction. (D) Gene expression of selected genes with a biphasic gene expression pattern, up early/down late, $n = 8$. DESeq2 was calculated using a Wald test with a Benjamini–Hochberg $p$-value correction. *$p < 0.05$ vs Pre values. Neg. negative, Reg. regulation.

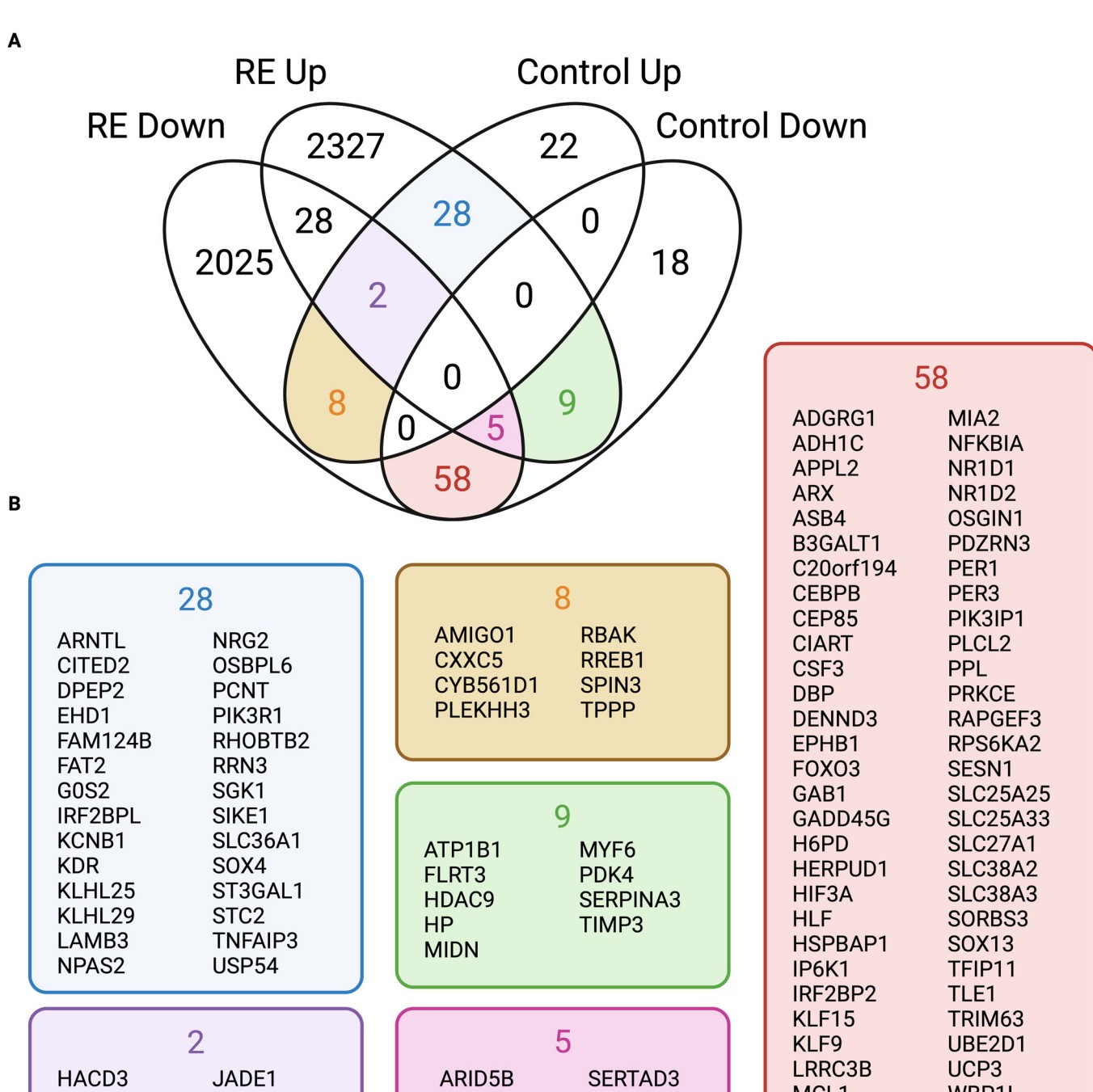

**Figure EV3. Venn diagram illustrating overlap in gene expression between the biopsy-only control group and the post-RE response at 3 and 8 h.**

(A) Venn-diagram of up and downregulated differentially expressed gene lists from the resistance exercise and control group. RE resistance exercise. (B) Genes corresponding to the overlap presented in (A).

**A**

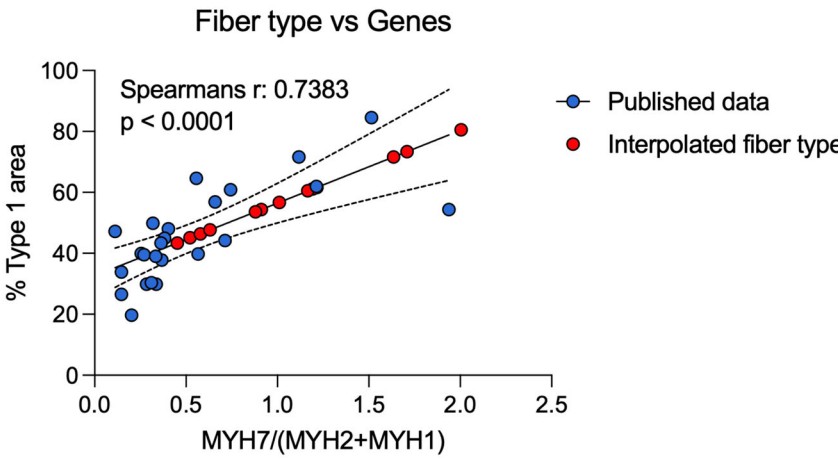

**B**

Interpolated fiber type area distribution

| Participant | Type I area % |
|-------------|---------------|
| Con 1 | 45.1 |
| Con 2 | 43.4 |
| Con 3 | 61.6 |
| Con 4 | 61.1 |
| Con 5 | 60.5 |
| Avg. | 54.3 ± 9.2 |
| RE 1 | 46.4 |
| RE 2 | 56.7 |
| RE 3 | 54.4 |
| RE 4 | 71.6 |
| RE 5 | 53.6 |
| RE 6 | 80.5 |
| RE 7 | 47.7 |
| RE 8 | 73.4 |
| Avg. | 60.5 ± 12.2 |

**Figure EV4. Interpolated fiber type area distribution of muscle samples.**

(A) Correlation of gene expression data with type I fiber area asses by muscle histology ($p = 3.0E-5$), data from Reitzner et al (2024). Blue dots = Data points from Reitzner et al (2024), Red dots = interpolated values based on gene data. (B) Data table of type I fiber area % in each participant.

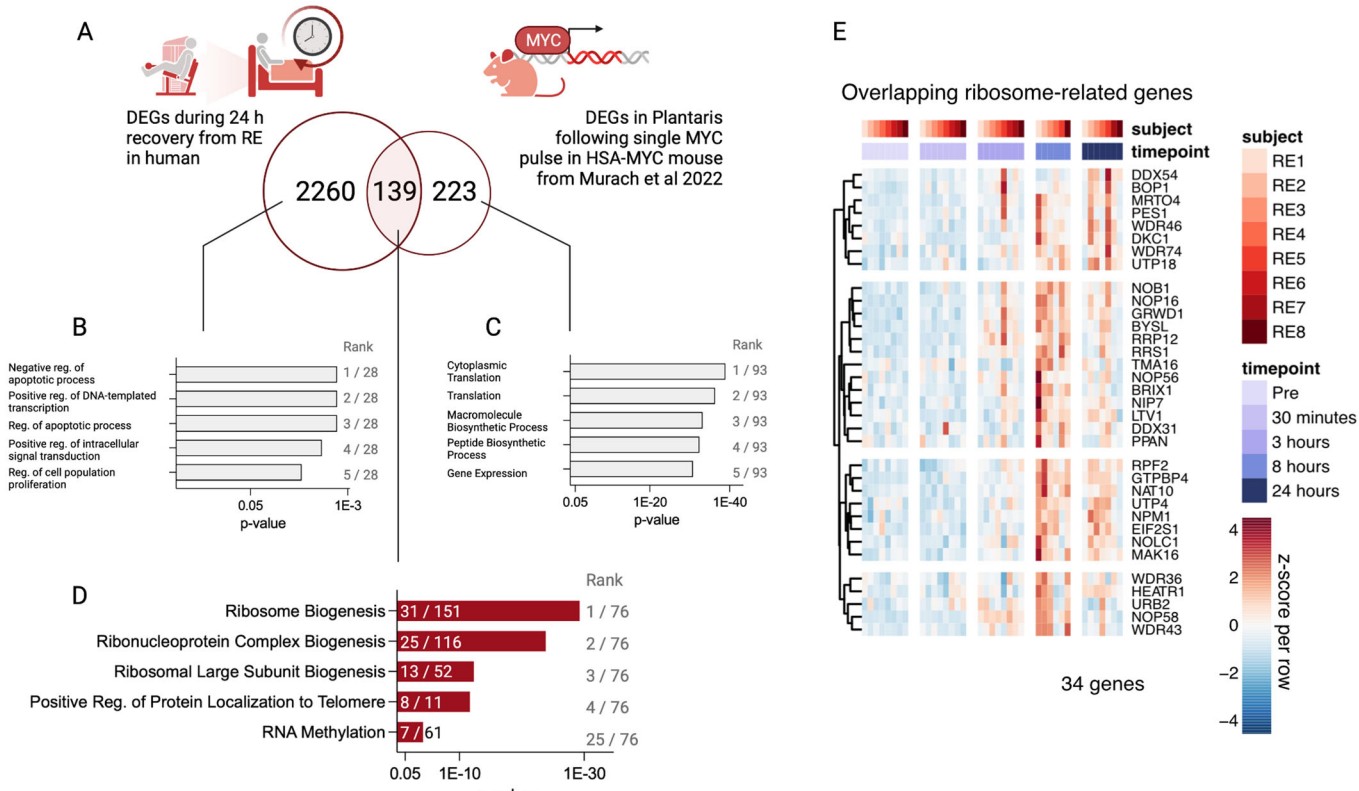

**Figure EV5. Transcriptional similarities between human RE recovery and MYC overexpression in mouse plantaris muscle.**

(A) Comparison of upregulated DEGs across 24 h of RE recovery in humans ($n = 8$) vs plantaris muscle from MYC-overexpressing mice from Murach et al (2022). (B–D) Top gene sets (GO: Biological processes) based on DEGs in (B) the human exclusive gene list, (C) MYC mouse exclusive gene list, and (D) overlapping gene list, respectively. (B–D) Gene ontology (GO) gene set enrichment analysis is analyzed using a Fisher exact test with Benjamini–Hochberg *p*-value correction. Gene sets are ranked according to their adj. *p*-values. (G) Heatmap showing DEG pattern for ribosome-related genes overlapping human RE response to a MYC response in mouse plantaris muscle. Source data are available online for this figure.

