## [Peer Review File · EMBO Reports]

The 24-Hour Molecular Landscape After Exercise in Humans Reveals MYC is Sufficient for Muscle Growth

Sebastian Edman, Ronald Jones III, Paulo Jannig, Rodrigo Fernandez-Gonzalo, Jessica Norrbom, Nicholas Thomas, Sabin Khadgi, Pieter Koopmans, Francielly Morena, Toby Chambers, Calvin Peterson, Logan Scott, Nicholas Greene, Vandre Figueiredo, Christopher Fry, Liu Zhengye, Johanna Lanner, Yuan Wen, Björn Alkner, Kevin Murach, and Ferdinand von Walden

Corresponding author(s): Kevin Murach (kmurach@uark.edu), Ferdinand von Walden (ferdinand.von.walden@ki.se)

Review Timeline:

Submission Date:	19th Apr 24
Editorial Decision:	5th Jun 24
Revision Received:	11th Sep 24
Editorial Decision:	2nd Oct 24
Revision Received:	8th Oct 24
Accepted:	16th Oct 24

Editor: Esther Schnapp

Transaction Report:

Dear Dr. Murach

Thank you for the submission of your manuscript to EMBO reports. We have now received the full set of referee reports that is pasted below.

As you will see, referees 1 and 3 acknowledge that the findings are potentially interesting, while referee 2 points out the somewhat limited novelty. All referees have several suggestions for how the study can be improved, and I think all should be addressed. Please let me know in case you disagree, and we can discuss this further, also in a video chat, if you like.

I would thus like to invite you to revise your manuscript with the understanding that the referee concerns must be fully addressed and their suggestions taken on board. Please address all referee concerns in a complete point-by-point response. Acceptance of the manuscript will depend on a positive outcome of a second round of review. It is EMBO reports policy to allow a single round of major revision only and acceptance or rejection of the manuscript will therefore depend on the completeness of your responses included in the next, final version of the manuscript.

We realize that it is difficult to revise to a specific deadline. In the interest of protecting the conceptual advance provided by the work, we recommend a revision within 3 months (5th Sep 2024). Please discuss the revision progress ahead of this time with the editor if you require more time to complete the revisions.

- 1) A data availability section providing access to data deposited in public databases is missing. If you have not deposited any data, please add a sentence to the data availability section that explains that.
- 2) Your manuscript contains statistics and error bars based on $n=2$. Please use scatter blots in these cases. No statistics should be calculated if $n=2$.

5) a complete author checklist, which you can download from our author guidelines <https://www.embopress.org/page/journal/14693178/authorguide>. Please insert information in the checklist that is also reflected in the manuscript. The completed author checklist will also be part of the RPF.

6) Please note that all corresponding authors are required to supply an ORCID ID for their name upon submission of a revised manuscript (<https://orcid.org/>). Please find instructions on how to link your ORCID ID to your account in our manuscript tracking system in our Author guidelines <https://www.embopress.org/page/journal/14693178/authorguide#authorshipguidelines>

7) Before submitting your revision, primary datasets produced in this study need to be deposited in an appropriate public

database (see <https://www.embopress.org/page/journal/14693178/authorguide#datadeposition>). Please remember to provide a reviewer password if the datasets are not yet public. The accession numbers and database should be listed in a formal "Data Availability" section placed after Materials & Method (see also <https://www.embopress.org/page/journal/14693178/authorguide#datadeposition>). Please note that the Data Availability Section is restricted to new primary data that are part of this study. * Note - All links should resolve to a page where the data can be accessed. *

I look forward to seeing a revised form of your manuscript when it is ready.

Esther Schnapp, PhD
Senior Editor

Referee #1:

In this manuscript Edman et al. investigated the how resistance exercise (RE) over a 24h timeframe alters the transcriptional output and the DNA methylation of human skeletal muscles, by analyzing muscle biopsies collected at different timepoints before and after exercise. The authors report that while changes in the control group reflected muscle circadian rhythms and food intake, RE activates ribosome biogenesis, actin folding, and autophagy. MYC was identified as a key driver of RE-induced changes in skeletal muscles. And selective expression of MYC in myofibers of transgenic mice partially recapitulated transcriptional changes observed upon RE in humans, with the notable fiber-type specific effects, whereby MYC-mediated increase in cross-sectional area of MyHC I-expressing fibers was restricted to the soleus muscle.

Despite the limited number of human subjects, this study provides in principle a valuable resource for the scientific community. The analysis of gene expression and DNA methylation patterns at sequential timepoints after exposure to RE significantly expands our insights into the mechanism underlying human skeletal myofiber hypertrophy. While the identification of MYC as key inducer of hypertrophy does not represent an absolute novelty, as it was reported in previous publications, this is the first study showing a similar role exerted by MYC in activating hypertrophy in both human and mouse muscles as well as the first study that implies MYC as terminal effector of the interplay between early alterations of DNA methylome and following induction of gene expression and translation into proteins.

I recommend that the authors address the points listed below:

- 1) Since all the analyses presented here were performed at the bulk level, it would be interesting to pinpoint the transcriptional changes found after RE to the various muscle resident cell types, through deconvolution strategies. A previous study of the effects of exercise in humans (see <https://doi.org/10.1038/s42003-022-04088-z>) at the single cell level has found that after 3h there is a shift in muscle-resident cells composition, due to an enrichment of infiltrating immune cells, which could partially explain why most of the transcriptional changes reported here are observed at later timepoints after RE.
- 2) As a follow-up of point 1, the authors might want determine whether myofibers provide the only source of MYC or whether additional cell type(s) express MYC following RE.
- 3) An interesting observation that is not pursued or discussed in the current work is the presence of PIEZO1 among the top up-regulated DEGs at 3h, as also predicted by the methylome at 30 minutes: PIEZO1 is an essential mediator of muscle stem cell (MuSCs) reactivity to changes in the microenvironment, plus it has been linked to endothelial cells response to exercise, leading to increased blood flow. Is PIEZO1 still upregulated at 24h post RE? Would this imply a more pro-activated MuSCs status in response to exercise? Can PIEZO1 promote MuSC activation toward an hypertrophic response induced by RE?
- 4) Why is the soleus the principal hindlimb muscle undergoing hypertrophy in response to MYC pulses? The authors should control whether this effect is due to different patterns of HSA expression in soleus vs other muscles?
- 5) Finally, it would be great if the authors could create a ShinyApp webpage of the human dataset's DESeq2 object, thus allowing unexperienced users to perform custom data comparisons and Gene Ontology analyses.

Referee #2:

This study by Edman et al investigates the transcriptional changes after a single bout of exercise in humans. Then they compare this to the changes occurring when increasing myc activity in mice in a pulsed manner. They observe a mild fiber hypertrophy in slow muscle fibers of the soleus after 4 weeks of pulsed myc activation.

The additional value of this lists of transcripts is somewhat limited after the recent MoTrPAC study in nature, and another similar analysis in acute exercise with a focus on pgc1-alpha (Furrer et al, 2023). These other studies take away some of the novelty of the first three figures of this manuscript.

- Some language use should be done in a simpler way here and there; phrases like 'The number of protein-coding differentially expressed genes (DEGs, adj. $p < 0.05$) during the time course of recovery (Figure 1A) in the repeated biopsies control group (CON, $n=5$) relative to the Pre time point (morning biopsy; Figure 1A) was' are not easy to read and should be written in a better way.
- It is not exactly clear what the message of figure 1 is. They see some changes in circadian genes at different timepoints in the control group. Since they see significant changes in PER genes, one would expect to see also changes in Bmal1, however this is not shown. Can they explain this? I ask, also because if there were actually major changes due to circadian gene regulation, one would expect to see bigger changes in the number of dysregulated genes.
- The interpretation of all these gene transcription changes would be helped if there was histology on the different samples and some info on the fiber type distribution in the sample.
- As only few genes follow the 'expected' methylation-down regulation scheme, this might not be such a clear correlation as is suggested in the text.
- The increase in fiber size after Myc activation is very mild and only occurring in slow fibers in the soleus. As this means the

growth response is highly specific and makes it critical to have data on the fiber type in the biopsies as well.

Referee #3:

This manuscript describes a well controlled study that seeks to further the understanding of the transcriptional response to a hypertrophic stimulus. The study involves a 24h time course in human skeletal muscle after acute resistance exercise. The study examined the relationship between changes in the DNA methylome and up regulated gene expression 3h post exercise, and the specific genes/pathways differentially regulated over 24h post-exercise and magnitude of these changes. Amongst the many findings was that MYC likely plays a major role in regulating post-exercise gene expression. Thus, the hypothesis was tested that MYC may be sufficient to induce muscle/muscle fiber hypertrophy. To this end, Myc expression was activated in a pulsatile manner in mice over 4wk, leading to increased whole muscle mass and fiber size in the slow-twitch soleus muscle but not fast-twitch muscles. Overall, the study provides important new insights into the transcriptional response in skeletal muscle in the post-resistance exercise recovery period. Also, in contrast to models of constitutive Myc overexpression in muscle, this study demonstrates that the transient upregulation of Myc may be a positive regulator of skeletal muscle mass and the adaptive hypertrophic response to resistance exercise. The manuscript is well written and presented.

My main comments (below) relate to the GO pathway enrichment analysis, the level of Myc protein overexpression over the 48 pulse period in mice compared to that typically observed in human muscle after resistance exercise, and potential muscle-specific differences in MYC protein levels in the 48h pulsing mouse model.

Comments:

Page 6, para 3, line 2- it is not clear what "background corrected" means here. This should be expanded on, perhaps in the Methods section.

Regarding the gene ontology and pathway analysis, which features prominently in this study, what specific background reference gene list was used? Was it a skeletal muscle-specific gene list or a generic 'cell' list (typically proliferative/cancer cells)? Clearly, this has no impact of the significant DEGS but can have a marked impact on the pathway analysis, with typically lower numbers of significant pathways when using a muscle-specific background. If a muscle-specific list was not used, please justify why one shouldn't be used? If the analysis program does not have a muscle background, then a custom background can be made using all the genes detected in your muscle tissue (regardless of whether they were significantly altered), and the DEGs can then be compared for enrichment against this background.

Page 7, para 4, line 4 - just a comment - it is interesting to note that of the genes listed in the ribosome biogenesis GO term, very few are for ribosomal proteins per se, but rather proteins involved in ribosome biogenesis/assembly. Perhaps this is consistent with a large number of ribosomal proteins being regulated at the translation level.

Page 8, para 3, line 14 - please add the specific references at the end of this statement.

Page 11, para 2, line 9 - the data for RUNX1 is not included in Fig. 4B, as indicated here.

Page 12, para 2, lines 1-4 - why use all upregulated genes in the analysis to determine which transcription factors might be regulating changes in ribosomal biogenesis and RNA-binding genes, and not just use the upregulated ribosomal biogenesis and RNA-binding genes themselves?

Page 13, para 3, lines 5-6 -data must be included to support the statement here that the level of Myc protein after 48h Dox is similar to Myc protein after several sessions of resistance exercise. This is an important point.

The issue of why pulsed Myc seems to have only impacted muscle mass/fiber size in the slow-twitch soleus muscle is not well addressed in the manuscript. Related to this, how does the Dox-induced increase in Myc protein differ between soleus, plantaris and Quad muscles? This should be included. Is the lack of response in the fast-twitch muscles due to a lower induction of Myc protein compared to soleus, or is there a lack of response despite similar a similar increase in Myc protein? This data, and some more discussion, would strengthen the manuscript.

Page 15, para 1, line 2-3 -- have the Authors shown an association here or demonstrated cause and effect? The phrase "strongly influences" seems to fit better with the demonstration of cause and effect.

Page 15, para 2, line 12 - provide some example references for this point.

Page 17, para 2, line 1 - How was this subset chosen? Randomly or based on some other parameter? This should be included.

Referee #1:

In this manuscript Edman et al. investigated the how resistance exercise (RE) over a 24h timeframe alters the transcriptional output and the DNA methylation of human skeletal muscles, by analyzing muscle biopsies collected at different timepoints before and after exercise. The authors report that while changes in the control group reflected muscle circadian rhythms and food intake, RE activates ribosome biogenesis, actin folding, and autophagy. MYC was identified as a key driver of RE-induced changes in skeletal muscles. And selective expression of MYC in myofibers of transgenic mice partially recapitulated transcriptional changes observed upon RE in humans, with the notable fiber-type specific effects, whereby MYC-mediated increase in cross-sectional area of MyHC I-expressing fibers was restricted to the soleus muscle.

Despite the limited number of human subjects, this study provides in principle a valuable resource for the scientific community. The analysis of gene expression and DNA methylation patterns at sequential timepoints after exposure to RE significantly expands our insights into the mechanism underlying human skeletal myofiber hypertrophy. While the identification of MYC as key inducer of hypertrophy does not represent an absolute novelty, as it was reported in previous publications, this is the first study showing a similar role exerted by MYC in activating hypertrophy in both human and mouse muscles as well as the first study that implies MYC as terminal effector of the interplay between early alterations of DNA methylome and following induction of gene expression and translation into proteins.

We thank the reviewer for their comments and for recognizing the importance of our manuscript.

I recommend that the authors address the points listed below:

1) Since all the analyses presented here were performed at the bulk level, it would be interesting to pinpoint the transcriptional changes found after RE to the various muscle resident cell types, through deconvolution strategies. A previous study of the effects of exercise in humans (see <https://doi.org/10.1038/s42003-022-04088-z>) at the single cell level has found that after 3h there is a shift in muscle-resident cells composition, due to an enrichment of infiltrating immune cells, which could partially explain why most of the transcriptional changes reported here are observed at later timepoints after RE.

We thank the reviewer for the suggestion. As suggested, we used the data set generated by Lovric et al. to run a deconvolution analysis. The resident cell types were estimated using the computational tool CIBERSORTx (PMID: 31061481). We estimated no changes in mononucleated cell proportions across the recovery period. We suspect that the difference in exercise modality (high-intensity interval training, HIIT, vs resistance exercise) may play a role.

The analysis has been included in the manuscript and the figure below is included as an Extended View figure.

“Over 90% of the transcriptomic signatures were estimated to originate from myofibers irrespective of time point. This proportion was estimated using CIBERSORTx⁶⁰, a computational cellular deconvolution tool that we have used previously for inferring changes in cell type after an acute hypertrophic stimulus²⁴. A recent acute exercise and single cell

RNA-seq study in human skeletal muscle was used as the reference dataset ⁶¹. The exercise involved intense cycle sprint intervals and the muscle sampled was the vastus lateralis, which corresponds with our study design. In this dataset ⁶¹, “myocytes” were inferred to be myonuclei based on the expression of adult myosin heavy chains and other muscle fiber-enriched markers. Myonuclei usually appear in skeletal muscle single cell datasets and are sequenced alongside mononuclear cells ^{62,63}. When excluding “myocytes” or “myofibers”, no appreciable change in mononucleated cell proportions was estimated throughout the recovery (Figure EV1).”

“Expanded View Figure 1. Cell composition of muscle biopsies through data deconvolution. Cell composition of all skeletal muscle biopsy time points excluding myocytes. SC= Satellite cell, CTRL = Control group, RE = Resistance exercise group.”

2) As a follow-up of point 1, the authors might want determine whether myofibers provide the only source of MYC or whether additional cell type(s) express MYC following RE.

It is possible that cell types other than muscle fibers contribute to MYC expression in skeletal muscle since MYC is induced during cellular proliferation, which surely occurs in mononuclear cells in muscle after exercise. In humans, however, the proportion of myonuclei versus nuclei from mononuclear cell types is very high (up to 90% myonuclei according to Orchard et al. and ~75% according to Eraslan et al):

<https://genome.cshlp.org/content/31/12/2258.full.pdf+html>

<https://www.science.org/doi/full/10.1126/science.abl4290>

Thus, the high levels of MYC we observe are likely derived in large part from the highly abundant myonuclei. Although we do not have tissue available to perform histology or additional single myonucleus sequencing experiments in the current study, there is very strong evidence from the literature – including work from our own

laboratory - that *Myc* is expressed specifically by myonuclei during skeletal muscle hypertrophy.

Using a skeletal muscle fiber myonucleus-specific labeling mouse, fluorescent myonuclear sorting, and a mechanical overload stimulus, we showed that *Myc* is robustly upregulated in myonuclei during a muscle hypertrophic stimulus in mice:

[https://www.jbc.org/article/S0021-9258\(22\)00958-9/fulltext](https://www.jbc.org/article/S0021-9258(22)00958-9/fulltext)

Similarly, work from Dr. Karyn Esser showed that MYC protein localized to myonuclei during mechanical overload in mice:

<https://journals.physiology.org/doi/full/10.1152/ajpcell.00093.2005>

Classic work from Dr. Stephen Always shows that MYC protein is localized to myonuclei during muscle hypertrophy in the quail:

<https://academic.oup.com/biomedgerontology/article/52A/4/B203/581501>

Finally, in other models of muscle stress such as acute ethanol ingestion, MYC protein only localizes to myonuclei in skeletal muscle (and not interstitial nuclei):

[https://www.metabolismjournal.com/article/S0026-0495\(02\)00080-X/fulltext](https://www.metabolismjournal.com/article/S0026-0495(02)00080-X/fulltext)

The phenotype we see with muscle fiber-specific MYC induction in our model is similar to what we see with our translatable exercise training model for mice (PoWeR) where *Myc* is also upregulated in the muscle (see Englund et al. 2021 *Function*). We are, therefore, confident that *MYC* expression in our human tissue after resistance training is driven by muscle fibers.

3) An interesting observation that is not pursued or discussed in the current work is the presence of PIEZO1 among the top up-regulated DEGs at 3h, as also predicted by the methylome at 30 minutes: PIEZO1 is an essential mediator of muscle stem cell (MuSCs) reactivity to changes in the microenvironment, plus it has been linked to endothelial cells response to exercise, leading to increased blood flow. Is PIEZO1 still upregulated at 24h post RE? Would this imply a more pro-activated MuSCs status in response to exercise? Can PIEZO1 promote MuSC activation toward an hypertrophic response induced by RE?

We want to thank the reviewer for this insightful comment. PIEZO1 is certainly a plausible key mediator of the RE response; however, in this dataset follow-up analysis revealed that the expression peaked at the 3h timepoint, then fell back at 8- and 24h, with fold changes at log₂FC 0.54 p=0.01 (3h), log₂FC 0.07 p=0.85 (8h), and log₂FC 0.11 p=0.80 (24h) for the three timepoints, respectively. Albeit still intriguing, we therefore choose not to include a section on PIEZO1 in the manuscript. However, any particular gene of interest will be easily accessible using our ShinyApp, (see response to comment 5 below).

4) Why is the soleus the principal hindlimb muscle undergoing hypertrophy in

response to MYC pulses? The authors should control whether this effect is due to different patterns of HSA expression in the soleus vs. other muscles.

This is a very important point. To ensure that the observed hypertrophy of the soleus muscle, as compared to other lower leg muscles, following repeated pulses of MYC over 4 weeks was not caused by differences in MYC induction, we performed western blots on Soleus, Gastrocnemius, Tibialis anterior, and Plantaris muscle for MYC following DOX treatment (new Figure 6B). Here, we see that all muscles, regardless of underlying fiber type, experience a robust increase of MYC. While MYC levels seem to be slightly elevated in soleus compared to Gastroc and TA, soleus and plantaris MYC levels are equal (even though soleus and plantaris have different growth outcomes from this treatment, as well as differences in gene expression after MYC induction as we have previously reported – see Murach et al. 2022 *JBC* versus Jones III et al. 2023 *J Physiol*). It is also noteworthy that although gastroc and TA has lower total MYC expression here, they are still over 100-fold increased from the control. We have added the following to the text in the last section of the Results:

“The doxycycline treatment caused MYC to be significantly induced in muscle specifically (Figure 6B, Appendix Figure S2 & S3). The 48-hour pulse strategy induced a similar amount of MYC protein in the soleus and plantaris muscles and a weaker induction in the gastrocnemius and tibialis anterior muscles; however, the induction was significant across all muscles. The administration pattern was chosen to approximate MYC induction in skeletal muscle by a regular RE regimen.”

“Figure 6. Four weeks of pulsed MYC induction is sufficient to elicit muscle fiber type specific hypertrophy. (B) Soleus, Gastrocnemius (Gastroc.), Tibialis anterior (TA), and Plantaris (Plant.) muscle stained for MYC after 48 Hours of doxycycline administration. + = HSA-MYC, - = HSA Control.....”

Collectively, these data suggest to us that the observed hypertrophy of specifically soleus may be an intrinsic effect of the more oxidative type I & IIa fibers of the soleus, and not related to the differences of MYC expression. What the underlying

mechanism is, however, is still unknown and will be the topic of future and ongoing investigations by our lab.

5) Finally, it would be great if the authors could create a ShinyApp webpage of the human dataset's DESeq2 object, thus allowing unexperienced users to perform custom data comparisons and Gene Ontology analyses.

We want to thank the reviewer for pointing out this opportunity. Based on this suggestion, we have created a ShinyApp to enable easier access to our data, and the link has now been incorporated into the manuscript as well. We hope, as the reviewer suggested, that this will enable more researchers to access our data.

<http://data.myoanalytics.com>

Referee #2:

This study by Edman et al investigates the transcriptional changes after a single bout of exercise in humans. Then they compare this to the changes occurring when increasing myc activity in mice in a pulsed manner. They observe a mild fiber hypertrophy in slow muscle fibers of the soleus after 4 weeks of pulsed myc activation.

The additional value of this lists of transcripts is somewhat limited after the recent MoTrPAC study in nature, and another similar analysis in acute exercise with a focus on pgc1-alpha (Furrer et al, 2023). These other studies take away some of the novelty of the first three figures of this manuscript.

We want to thank the reviewer for taking the time to thoroughly read through our manuscript, and providing helpful comments to increase the readability. While we fully acknowledge the massive impact that the recent MoTrPAC study will have on the field, it is our position that the coexistence with the MoTrPAC-study (or the study by Furrer et al.) does not detract from the novelty of our work. Instead, we suggest that these papers are synergistic. What our paper adds (specifically with reference to the first three figures) is a high-resolution timeline of transcriptional changes to resistance exercise in a human model. The MoTrPAC study and Furrer et al.'s work are both specifically related to endurance exercise, are performed in pre-clinical models, and lack the temporal resolution that we provide in human samples. All biopsy time points (5 biopsies per individual; Pre, 30min post, 3h post, 8h post, and 24h post) are taken in the same individual, further strengthening our design, in addition to the biopsy-only control group. For these reasons, we feel strongly that our work will be impactful and well-cited, alongside the other studies mentioned.

- Some language use should be done in a simpler way here and there; phrases like 'The number of protein-coding differentially expressed genes (DEGs, adj. $p < 0.05$) during the time course of recovery (Figure 1A) in the repeated biopsies control group (CON, $n=5$) relative to the Pre time point (morning biopsy; Figure 1A) was' are not easy to read and should be written in a better way.

We have tried to simplify the writing in some sections of the manuscript. For instance, the section mentioned now reads:

"..Differentially expressed genes (DEGs, adj. $p < 0.05$) were analyzed relative to the collected Pre biopsy. In the control group (CON, $n=5$) the number of DEGs at the different recovery time points was:.."

Hopefully, this improves readability without losing the critical information that we want to convey with the section.

- It is not exactly clear what the message of figure 1 is. They see some changes in circadian genes at different timepoints in the control group. Since they see significant changes in PER genes, one would expect to see also changes in Bmal1, however this is not shown. Can they explain this? I ask, also because if there were actually major changes due to circadian gene regulation, one would expect to see bigger changes in the number of dysregulated genes.

The general message of Figure 1 is partly to show that our repeated biopsy procedure, sampling five muscle biopsies from vastus lateralis within 24 hours does not interfere with what we observe in the resistance exercise (RE) time course later presented. For instance, we show that the few genes that are being differentially expressed are related to circadian rhythmicity (Figure 1C, selection of highlighted genes) and that the genes collectively are not significantly skewed towards pathways that could interfere with our results in the RE-trial.

In our data set, *BMAL1*, also referred to as *ARNTL*, is significantly changed at our 8h timepoint, log2FC 2.2, p=0.01. We, therefore, added ARNTL to figure 1C where we highlight certain genes connected to circadian rhythmicity.

Finally, as per the suggestion of the referee #1, we have created a ShinyApp available online (<http://data.myoanalytics.com>). Here, readers will be able to look up any specific genes of interest, both in the RE and control group, thereby making our rich dataset more accessible

- The interpretation of all these gene transcription changes would be helped if there was histology on the different samples and some info on the fiber type distribution in the sample.

Since we do not have muscle samples prepared for histology, we are unable to conduct such an experiment. However, to circumvent this and address the reviewer's question about fiber type, we added a figure in which we interpolate fiber type distribution in our samples using ratios of myosin transcripts, and added a section on inferring fiber type from transcriptional data

“Inferring fiber type distribution from the transcriptional data across cohorts

To characterize the fiber type distribution of the RE and CON group, we used the gene counts of adult myosin heavy chain mRNAs (MYH1 – Type IIX, MYH2 – Type IIA, and MYH7 – Type I) to interpolate skeletal muscle fiber type distribution. We accomplished this by leveraging a publicly available dataset that contained both transcriptional data and fiber type distribution using muscle histology⁷¹. Correlating the % Type I fiber area from each individual to the MYH transcript ratio (MYH7/(MYH2+MYH1)) yielded a strong significant correlation of $r=0.7178$ ($p<0.0001$, Figure EV 4A). We then used these values to create a standard curve and fitted our MYH-transcript data along the fitted line, thus estimating a fiber-type distribution in our samples. The muscle samples from the CON group and RE group consisted of 54.3 ± 9.2 and 60.5 ± 12.2 % type I fiber area (Figure EV 4B), respectively, with no difference between the two groups ($p=0.33$).”

A**B**

Interpolated fiber type area distribution

Participant	Type I area %
Con 1	45.1
Con 2	43.4
Con 3	61.6
Con 4	61.1
Con 5	60.5
Avg.	54.3 ± 9.2
RE 1	46.4
RE 2	56.7
RE 3	54.4
RE 4	71.6
RE 5	53.6
RE 6	80.5
RE 7	47.7
RE 8	73.4
Avg.	60.5 ± 12.2

Expanded View Figure 4. Interpolated fiber type area distribution of muscle samples.

(A) Correlation of gene expression data with type I fiber area asses by muscle histology, data from Reitzner et al. (2024). Blue dots = Data points from Reitzner et al. (2024), Red dots = Interpolated values based on gene data. (B) Data table of type I fiber area % in each participant.

- As only few genes follow the 'expected' methylation-down regulation scheme, this might not be such a clear correlation as is suggested in the text.

We agree, and have added the following to the text in order to emphasize this point:

“This analysis [BETA] inferred significant methylation control of 936 upregulated genes at 3 hours ($p=0.000007$), and 805 downregulated genes were identified according to BETA ($p<0.05$), but the overall regulation of repressed genes was not significant according to the Kolmogorov-Smirnov test ($p=0.952$). It is important to note that the lack of significance according to BETA for downregulated genes does not mean that methylation is not regulating gene expression on a gene-by-gene basis, but that the global regulatory potential score did not achieve significance. Thus, we present the BETA analysis for individual genes to provide additional insights.”

- The increase in fiber size after Myc activation is very mild and only occurring in slow fibers in the soleus. As this means the growth response is highly specific and makes it critical to have data on the fiber type in the biopsies as well.

The hypertrophy seen with pulsatile MYC induction over four weeks is similar to what we observe with PoWeR training over four weeks in the soleus (see Englund et al. 2021 *Function*). Furthermore, our pulsatile MYC findings are translatable because almost all other gain of function studies ever performed in skeletal muscle involve chronic/constitutive induction – which is arguably non-physiological – and tends to focus on MyHC IIB-containing muscles, which is a fiber type that humans do not express. The Soleus muscle of the mouse is quite similar in fiber type to the human Vastus Lateralis (~50/50 MyHC 1 vs 2).

To provide insight on the contribution of fiber type distribution to MYC induction, we have added the following to the final paragraph of the Results:

“Given the fiber type-specific effects of Myc induction seen in the current, and previous work, we revisited our human time course data (Figure 1-2), asking if the degree of MYC expression could be related to fiber type distribution. However, no such indications were found, with peak MYC expression at 3- and 8h (Figure 2C) showing correlations of $r=0.37$ ($p=0.29$; Spearman) and $r=-0.31$ ($p=0.38$; Spearman) vs type I fiber distribution, respectively.”

Referee #3:

This manuscript describes a well controlled study that seeks to further the understanding of the transcriptional response to a hypertrophic stimulus. The study involves a 24h time course in human skeletal muscle after acute resistance exercise. The study examined the relationship between changes in the DNA methylome and up regulated gene expression 3h post exercise, and the specific genes/pathways differentially regulated over 24h post-exercise and magnitude of these changes. Amongst the many findings was that MYC likely plays a major role in regulating post-exercise gene expression. Thus, the hypothesis was tested that MYC may be sufficient to induce muscle/muscle fiber hypertrophy. To this end, Myc expression was activated in a pulsatile manner in mice over 4wk, leading to increased whole muscle mass and fiber size in the slow-twitch soleus muscle but not fast-twitch muscles. Overall, the study provides important new insights into the transcriptional response in skeletal muscle in the post-resistance exercise recovery period. Also, in contrast to models of constitutive Myc overexpression in muscle, this study demonstrates that the transient upregulation of Myc may be a positive regulator of skeletal muscle mass and the adaptive hypertrophic response to resistance exercise. The manuscript is well written and presented.

We thank the reviewer for their insightful comments and appreciation of our manuscript.

My main comments (below) relate to the GO pathway enrichment analysis, the level of Myc protein overexpression over the 48 pulse period in mice compared to that typically observed in human muscle after resistance exercise, and potential muscle-specific differences in MYC protein levels in the 48h pulsing mouse model.

Comments:

Page 6, para 3, line 2- it is not clear what "background corrected" means here. This should be expanded on, perhaps in the Methods section.

In the revised version of the manuscript, we have expanded and further clarified the method section on how the background was set for the pathway analysis. However, in the results section, we have kept the original wording while citing Stokes et al. (2023; PMID: 37288167), who suggested this method for proper background correction. The method section, by contrast, now reads:

"Gene set enrichment analyses were conducted on the collapsed gene lists using Enrichr (<https://maayanlab.cloud/Enrichr/> 2023-Aug-16) with the 2023 gene ontology (GO) database as our cross reference⁵⁶⁻⁵⁹. We used all protein-coding genes detected in our muscle samples (14,392 genes, EV Table 3) as our background correction for the pathway analysis, as suggested by Stokes et al.⁵⁵."

Regarding the gene ontology and pathway analysis, which features prominently in this study, what specific background reference gene list was used? Was it a skeletal muscle- specific gene list or a generic 'cell' list (typically proliferative/cancer cells)? Clearly, this has no impact of the significant DEGS but can have a marked impact on the pathway analysis, with typically lower numbers of significant pathways when using a muscle- specific background. If a muscle-specific list was not used, please

justify why one shouldn't be used? If the analysis program does not have a muscle background, then a custom background can be made using all the genes detected in your muscle tissue (regardless of whether they were significantly altered), and the DEGs can then be compared for enrichment against this background.

This was addressed above, but to further clarify: As suggested by the reviewer, we used all protein-coding genes detected in our sequencing as the background for our pathway analysis. By using our own gene list, we will get a background that adjusts, both for the sensitivity of the sequencing itself and also the type of tissue input (i.e., skeletal muscle biopsies in this case).

Page 7, para 4, line 4 - just a comment - it is interesting to note that of the genes listed in the ribosome biogenesis GO term, very few are for ribosomal proteins per se, but rather proteins involved in ribosome biogenesis/assembly. Perhaps this is consistent with a large number of ribosomal proteins being regulated at the translation level.

This is an astute observation that we now highlight more clearly in the Results:

“Consequently, using the same gene set enrichment analysis on the 316 genes overlapping the human RE response and soleus transcriptome from the MYC overexpression data generated gene sets largely related to the ribosome (Figure 5F-H; Appendix Table S14) – specifically, genes coding proteins involved in ribosome biogenesis, assembly, and translation initiation and elongation (e.g. *EEF* and *EIF* genes).”

Page 8, para 3, line 14 - please add the specific references at the end of this statement.

This has been added to the manuscript.

Page 11, para 2, line 9 - the data for *RUNX1* is not included in Fig. 4B, as indicated here.

Thank you for pointing out this oversight to us. We have now adjusted the figure to include *RUNX1*.

Page 12, para 2, lines 1-4 - why use all upregulated genes in the analysis to determine which transcription factors might be regulating changes in ribosomal biogenesis and RNA-binding genes, and not just use the upregulated ribosomal biogenesis and RNA-binding genes themselves?

This is a good point, and we considered the option of being more specific in the LISA-analysis, as the reviewer suggested. We decided on the current form as we felt it would add the potential for a broader perspective, while not losing (too much of) the ribosome/myc focus.

Basically, running the Lisa analysis on the genes of interest (Ribo-related genes) would generate a very specific set of transcription factors for these pathways, where MYC would still be dominant. We felt that this would convey the same specific message as what we now are conveying, but with a less broad perspective on what

other processes and transcription factors may be important within these time points. Presenting the data timepoint specific, rather than ribosome/RNA specific, also helped transition the manuscript from the early timecourse focus, towards a MYC focus.

Nevertheless, both approaches to presenting the data are, of course, valid. Therefore, we also included a figure as a supplementary figure in the manuscript (Appendix figure S1) showing the transcription factors that are predicted to regulate the genes included in GO-pathways Ribosome Biogenesis and RNA-binding.

“Appendix Figure S1. Transcription factors associated with the most highly enriched Gene Ontology pathways. (A) Top 5 transcription factors predicted to be associated with genes regulating Ribosomal Biogenesis in our data set. (B) Top 5 transcription factors predicted to be associated with genes regulating RNA-binding in our data set.”

Page 13, para 3, lines 5-6 -data must be included to support the statement here that the level of Myc protein after 48h Dox is similar to Myc protein after several sessions of resistance exercise. This is an important point.

We thank the reviewer for pointing out that the wording used in our first version of the manuscript could be easily misinterpreted. What we meant to convey with this sentence was that our administration pattern of Dox was chosen to mimic the periodicity of regular exercise regimens. While some previously published data support the notion that MYC protein is upregulated at 24H following a single RE-session (Figueredo et al. 2021), our intentions were not to convey that our attained levels of MYC expression in our transgenic mouse model are a perfectly accurate representation of the human physiology. To emphasize this to further readers, the sentence in the manuscript has been changed to:

“The administration pattern was chosen to approximate MYC induction in skeletal muscle by a regular weekly RE regimen.”

Further, we have added a western blot to support the magnitude of MYC-induction in our model, see point below.

All of that is to say that, our pulsatile model is still arguably more translatable than chronic/constitutive induction models. What is more, studies in humans

(<https://journals.physiology.org/doi/full/10.1152/ajpendo.00486.2015>) and mice (<https://www.sciencedirect.com/science/article/pii/S0014579315003051>) qualitatively show similar inductions of MYC protein with exercise and chronic mechanical overload. The latter study (Goodman et al.) shows saturating levels of MYC via Western Blot in muscle during mechanical overload in mice, and we do not observe this same level of extreme induction with our MYC model.

The issue of why pulsed Myc seems to have only impacted muscle mass/fiber size in the slow-twitch soleus muscle is not well addressed in the manuscript. Related to this, how does the Dox-induced increase in Myc protein differ between soleus, plantaris and Quad muscles? This should be included. Is the lack of response in the fast-twitch muscles due to a lower induction of Myc protein compared to soleus, or is there a lack of response despite similar a similar increase in Myc protein? This data, and some more discussion, would strengthen the manuscript.

We have addressed this by adding an additional experiment (Figure 6B). We observe a robust increase in MYC in all examined muscles after can acute pulse of MYC (48 hours of induction followed by a 12 hour washout), with over 100-fold increases from control mice (MYC levels are generally undetectable in control mice). However, we did observe a varied MYC-expression in the lower leg following Dox treatment. This said, the soleus and plantaris muscles showed equally high MYC induction, yet this resulted in hypertrophic growth of the soleus while the plantaris remained unaffected. This leads us to believe that the soleus-specific growth is not due to differences in MYC induction between muscles but rather something intrinsic to the slow-twitch fibers. In context with our prior work evaluating the transcriptome in response to a pulse of MYC, these data suggest that the Soleus is more sensitive to MYC than the plantaris muscle (see Murach et al. JBC 2022 vs Jones et al. J Physiol 2023).

We have added the following to the text in the last section of the Results:

“The doxycycline treatment caused MYC to be significantly induced in muscle specifically (Figure 6B, Appendix Figure S1). The 48-hour pulse strategy induced a similar amount of MYC protein in the soleus and plantaris muscles and a weaker induction in the gastrocnemius and tibialis anterior muscles; however, the induction was significant across all muscles. The administration pattern was chosen to approximate MYC induction in skeletal muscle by a once-a-week RE regimen.”

And:

“Given the western blot data presented above across muscle groups, we infer that the soleus muscle is more sensitive to MYC induction than other muscles, specifically the plantaris. These differences in gene expression between muscles likely contributed to soleus-specific mass gains.”

“Figure 6. Four weeks of pulsed MYC induction is sufficient to elicit muscle fiber type specific hypertrophy. (B) Soleus, Gastrocnemius (Gastroc.), Tibialis anterior (TA), and Plantaris (Plant.) muscle stained for MYC after 48 Hours of doxycycline administration. + = HSA-MYC, - = HSA Control.....”

Page 15, para 1, line 2-3 -- have the Authors shown an association here or demonstrated cause and effect? The phrase "strongly influences" seems to fit better with the demonstration of cause and effect.

We have softened the language here, the line now reads:

“the DNA methylome response to RE at 30 minutes clearly associated with global gene expression at 3 hours”

Page 15, para 2, line 12 - provide some example references for this point.

We have added two references on NR4A3 and its potential for a biphasic expression pattern following exercise.

Page 17, para 2, line 1 - How was this subset chosen? Randomly or based on some other parameter? This should be included.

Thank you, this is important to highlight. The subset of volunteers chosen were individuals for whom we had the most complete biopsy sets available. This has been specified in the manuscript.

Dear Dr. Murach

Thank you for the submission of your manuscript to EMBO reports. We have now received the full set of referee reports that is pasted below.

As you will see, referees 1 and 3 acknowledge that the findings are potentially interesting, while referee 2 points out the somewhat limited novelty. All referees have several suggestions for how the study can be improved, and I think all should be addressed. Please let me know in case you disagree, and we can discuss this further, also in a video chat, if you like.

I would thus like to invite you to revise your manuscript with the understanding that the referee concerns must be fully addressed and their suggestions taken on board. Please address all referee concerns in a complete point-by-point response.

Acceptance of the manuscript will depend on a positive outcome of a second round of review. It is EMBO reports policy to allow a single round of major revision only and acceptance or rejection of the manuscript will therefore depend on the completeness of your responses included in the next, final version of the manuscript.

We realize that it is difficult to revise to a specific deadline. In the interest of protecting the conceptual advance provided by the work, we recommend a revision within 3 months (5th Sep 2024). Please discuss the revision progress ahead of this time with the editor if you require more time to complete the revisions.

1) A data availability section providing access to data deposited in public databases is missing. If you have not deposited any data, please add a sentence to the data availability section that explains that.

RNA-seq data is uploaded with tokens for reviewers. Source data is provided according to the request of SourceData Coordinator Hannah Sonntag, PhD.

2) Your manuscript contains statistics and error bars based on $n=2$. Please use scatter blots in these cases. No statistics should be calculated if $n=2$. When submitting your revised manuscript, please carefully review the instructions that follow below. Failure to include requested items will delay the evaluation of your revision.

No such error bars or statistical calculations is included in the manuscript.

This has been included in the submission of our revised manuscript. One clean version of the manuscript and a separate document where the major changes made to the manuscript are marked in red.

2) individual production quality figure files as .eps, .tif, .jpg (one file per figure). See

https://wol-prod-cdn.literatumonline.com/pb-assets/embo-site/EMBOPress_Figure_Guidelines_061115-1561436025777.pdf for more info on how to prepare your figures.

Figures have been included as individual .tif-files, as well as a multi-figure PDF file for the reviewers.

3) We replaced Supplementary Information with Expanded View (EV) Figures and Tables that are collapsible/expandable online. A maximum of 5 EV Figures can be typeset. EV Figures should be cited as 'Figure EV1, Figure EV2' etc... in the text and their respective legends should be included in the main text after the legends of regular figures.

<<https://www.embopress.org/page/journal/14693178/authorguide#expandedview>>;

- Additional Tables/Datasets should be labeled and referred to as Table EV1, Dataset EV1, etc. Legends have to be provided in a separate tab in case of .xls files.

Alternatively, the legend can be supplied as a separate text file (README) and zipped together with the Table/Dataset file.

This has been adjusted in the current version of the manuscript.

A separate .docx file has been included where we address the reviewers concerns.

5) a complete author checklist, which you can download from our author guidelines <<https://www.embopress.org/page/journal/14693178/authorguide>>;. Please insert information in the checklist that is also reflected in the manuscript. The completed author checklist will also be part of the RPF.

An author checklist has been provided with the second submission.

6) Please note that all corresponding authors are required to supply an ORCID ID for their name upon submission of a revised manuscript (<<https://orcid.org/>>). Please find instructions on how to link your ORCID ID to your account in our manuscript tracking system in our Author guidelines

<<https://www.embopress.org/page/journal/14693178/authorguide#authorshipguideline>>;

Sebastian Edman 0000-0003-2921-833X

Kevin A Murach 0000-0003-2783-7137

Ferdinand von Walden 0000-0003-1134-2252

7) Before submitting your revision, primary datasets produced in this study need to be deposited in an appropriate public database (see

<https://www.embopress.org/page/journal/14693178/authorguide#datadeposition>).

The accession numbers and database should be listed in a formal "Data Availability"

section placed after Materials & Method (see also <https://www.embopress.org/page/journal/14693178/authorguide#datadeposition>). Please note that the Data Availability Section is restricted to new primary data that are part of this study. * Note - All links should resolve to a page where the data can be accessed. *

Data has been deposited in GEO, and tokens have been made available for reviewers. Data will be made publicly available upon acceptance of the paper.

Source data is provided according to the request of SourceData Coordinator Hannah Sonntag, PhD.

9) Our journal also encourages inclusion of *data citations in the reference list* to directly cite datasets that were re-used and obtained from public databases. Data citations in the article text are distinct from normal bibliographical citations and should directly link to the database records from which the data can be accessed. In the main text, data citations are formatted as follows: "Data ref: Smith et al, 2001" or "Data ref: NCBI Sequence Read Archive PRJNA342805, 2017". In the Reference list, data citations must be labeled with "[DATASET]". A data reference must provide the database name, accession number/identifiers and a resolvable link to the landing page from which the data can be accessed at the end of the reference. Further instructions are available at

<https://www.embopress.org/page/journal/14693178/authorguide#referencesformat>
Data refs have been added to figure legends corresponding to Figure 5 and Figure EV 5.

10) Regarding data quantification (see Figure Legends:

<https://www.embopress.org/page/journal/14693178/authorguide#figureformat>)

The test applied has been added to the figure legends. P values and N number is included.

This is also included.

A conflict-of-interest statement has been included at the end of the manuscript.

A cover image suggestion has been included in the submission. We have hired a scientific illustrator to help us come up with a draft, and we would very much like to work with the Graphics Illustrators at EMBOpress in designing a cover for EMBO reports.

I look forward to seeing a revised form of your manuscript when it is ready.

Esther Schnapp, PhD Senior Editor

EMBO reports

Dear Dr. Murach,

Thank you for the submission of your revised manuscript. We have now received the enclosed reports from the referees and I am happy to say that all support its publication now. Please address all final referee comments in the next version of the ms.

Also a few editorial requests will need to be addressed before we can proceed with the official acceptance of your manuscript:

- Please note that the specific URL for the GSE252357 dataset needs to be provided in the data availability statement. This statement also needs to be placed before the Acknowledgments.
- The conflict of interest subheading needs to be corrected to "Disclosure Statement and Competing Interests"
- The author credits need to be removed from the ms file.
- This funding info is missing in our online submission system: the Arkansas Integrated Metabolic Research Center - AIMRC (P20GM139768), please add.
- A callout for Figure 6M is missing, please add.
- The excel file titled "Expanded View Tables" seems to contain a mix of Datasets and may be source data? Please upload all Datasets as separate excel Dataset files, called Dataset EV1, etc. A Dataset can have multiple sheets if the data are related (this may be relevant for Table 7A-C, Table 9A-B?). Each legend should be in a separate sheet or tab. Please also update all ms callouts accordingly. If some of the data are source data please upload these as source data (SD) files, 1 SD file per main ms figure.
- Please combine Appendix Figures S1-S3 and their legends in an Appendix PDF file; the file should have a table of content with page numbers and the legends should be removed from the ms file.
- Since July this year all ms need to have a Reagents and Tools Table (listing key reagents, experimental models, software and relevant equipment and including their sources and relevant identifiers). A downloadable templates (.docx) for the Reagents and Tools Table can be found in our author guidelines: <<https://www.embopress.org/page/journal/14693178/authorguide#manuscriptpreparation>>.
- The source data need to be uploaded as one zipped SD folder per main figure.
- Please write about your findings in the abstract in present tense, as per journal policy.

Please address these comments on the figure legends:

- Please note that the exact p values are not provided in the legends of figures 1c; 6b-d, j-l; EV 2d; EV 4a.
- Please indicate the statistical test used for data analysis in the legends of figures 1b-c, e-g; 2b; 3a-c; 4b-d; 5a-b, d-f; 6k-l; EV 2a-d; EV 5b-d.
- Please note that the box plots need to be defined in terms of minima, maxima, centre, bounds of box and whiskers, and percentile in the legends of figures 6k-m; EV 1.
- Please note that information related to n is missing in the legends of figures 1c; 2c; EV 1; EV 2d.
- Please note that the error bars are not defined in the legends of figures 1c; 2c; 6b-f, h-j; EV 2d.

- Please note that the ""Data ref: Jones et al (2023)"" and ""Data ref: Murach et al (2022)"" data citation does not refer to deposited experimental data.
- Please note that the ""Data ref: Jones et al (2023)"" and ""Data ref: Murach et al (2022)"" data citations are not tagged with the label "DATASET" in the reference list.
- Please note that the URL for ""Data ref: Jones et al (2023)"" and ""Data ref: Murach et al (2022)"" data citation is not provided.

The synopsis image you sent is good, but we also need a short (1-2 sentences) summary of the findings and their significance, and 2-3 bullet points highlighting key results from you, for our website.

Referee #1:

The reviewer is satisfied with the supplemental data generated and recommends the article for publication. Just remember to add the online resource (<http://data.myoanalytics.com/>) also to the Data Availability section.

Referee #2:

The authors present a much revised version of their manuscript. The effort to address the different issues is appreciated. It remains a shame there is no histological data on these samples, as this would have greatly improved the value of their story in my mind, they addressed the fiber type issue (a very substantial one) in a satisfactory manner. Overall I feel the manuscript as currently presented will add value to the community.

Minor remaining issue:

- As there is no statistical correlation between the methylome and the transcriptome, on a whole, also the abstract needs to be modified to underline this aspect. The same should be mentioned also in the beginning of the discussion

Referee #3:

The Authors are to be commended for their thoughtful responses and amendments to the manuscript, that have ultimately strengthened the work.
I have no further comments.

Dear Dr. Murach,

Thank you for the submission of your revised manuscript. We have now received the enclosed reports from the referees and I am happy to say that all support its publication now. Please address all final referee comments in the next version of the ms.

Also a few editorial requests will need to be addressed before we can proceed with the official acceptance of your manuscript:

- Please note that the specific URL for the GSE252357 dataset needs to be provided in the data availability statement. This statement also needs to be placed before the Acknowledgments.

The data availability section has been moved before acknowledgments, and the GSE URL is now incorporated into the section.

- The conflict of interest subheading needs to be corrected to "Disclosure Statement and Competing Interests"

The subheading now reads "Disclosure Statement and Competing Interests"

- The author credits need to be removed from the ms file.

Author credits have been removed.

- This funding info is missing in our online submission system: the Arkansas Integrated Metabolic Research Center - AIMRC (P20GM139768), please add.

This has been added to the online system.

- A callout for Figure 6M is missing, please add.

This has been added to the manuscript, along with adjustments of some callout-errors related to Figure 6.

- The excel file titled "Expanded View Tables" seems to contain a mix of Datasets and may be source data? Please upload all Datasets as separate excel Dataset files, called Dataset EV1, etc. A Dataset can have multiple sheets if the data are related (this may be relevant for Table 7A-C, Table 9A-B?). Each legend should be in a separate sheet or tab. Please also update all ms callouts accordingly. If some of the data are source data please upload these as source data (SD) files, 1 SD file per main ms figure.

The file Expanded View tables have now been split into smaller Excel files, with some named Dataset EVs. Most of the data sets have been moved into Source data.

- Please combine Appendix Figures S1-S3 and their legends in an Appendix PDF

file; the file should have a table of content with page numbers and the legends should be removed from the ms file.

A separate Appendix Figure PDF file has been created with a table of contents. The appendix information has been removed from the main ms file.

- Since July this year all ms need to have a Reagents and Tools Table (listing key reagents, experimental models, software and relevant equipment and including their sources and relevant identifiers). A downloadable templates (.docx) for the Reagents and Tools Table can be found in our author guidelines:

< <https://www.embopress.org/page/journal/14693178/authorguide#manuscriptpreparation>>.

A Reagents and Tools Table has been added to the submission

- The source data need to be uploaded as one zipped SD folder per main figure.

This has been adjusted in the file-upload

- Please write about your findings in the abstract in present tense, as per journal policy.

The abstract now reads in the present tense.

Please address these comments on the figure legends:

- Please note that the exact p values are not provided in the legends of figures 1c; 6b-d, j-l; EV 2d; EV 4a.

This has been added to the manuscript.

- Please indicate the statistical test used for data analysis in the legends of figures 1b-c, e-g; 2b; 3a-c; 4b-d; 5a-b, d-f; 6k-l; EV 2a-d; EV 5b-d.

This has been added to the manuscript.

- Please note that the box plots need to be defined in terms of minima, maxima, centre, bounds of box and whiskers, and percentile in the legends of figures 6k-m; EV 1.

This has been added to the manuscript.

- Please note that information related to n is missing in the legends of figures 1c; 2c; EV 1; EV 2d.

This has been added to the manuscript.

- Please note that the error bars are not defined in the legends of figures 1c; 2c; 6b-f, h-j; EV 2d.

This has been added to the manuscript. Error bars have been added to bar charts in figure 6.

- Please note that the ""Data ref: Jones et al (2023)"" and ""Data ref: Murach et al (2022)"" data citation does not refer to deposited experimental data.
 - Please note that the ""Data ref: Jones et al (2023)"" and ""Data ref: Murach et al (2022)"" data citations are not tagged with the label "DATASET" in the reference list.
 - Please note that the URL for ""Data ref: Jones et al (2023)"" and ""Data ref: Murach et al (2022)"" data citation is not provided.
- Data ref was not appropriate as data was taken directly from journal appendix files, not a database. Data refs have instead been “converted” to regular references.

The data citation in the manuscript was not a proper use of this function and has thus been exchanged to a regular reference.

The synopsis image you sent is good, but we also need a short (1-2 sentences) summary of the findings and their significance, and 2-3 bullet points highlighting key results from you, for our website.

The synopsis and bullet points have been removed from the manuscript and provided in a separate file.

Referee #1:

The reviewer is satisfied with the supplemental data generated and recommends the article for publication. Just remember to add the online resource (<http://data.myoanalytics.com/>) also to the Data Availability section.

We thank the reviewer for improving the quality of the manuscript and helping with the accessibility of our data.

The online resource link has been added to the Data Availability section.

Referee #2:

The authors present a much revised version of their manuscript. The effort to address the different issues is appreciated. It remains a shame there is no

histological data on these samples, as this would have greatly improved the value of their story in my mind, they addressed the fiber type issue (a very substantial one) in a satisfactory manner. Overall I feel the manuscript as currently presented will add value to the community.

Minor remaining issue:

- As there is no statistical correlation between the methylome and the transcriptome, on a whole, also the abstract needs to be modified to underline this aspect. The same should be mentioned also in the beginning of the discussion

We want to thank the referee for taking the time to thoroughly review our manuscript. We are in agreement with the referee that preparing muscle for histology could have strengthened the manuscript. Regardless, we are happy to read the reviewer's assessment and agree that we feel the data presented herein will add value to the field.

Regarding the final issue, we have added the following description to the Results to further describe the BETA approach, which is what we used instead of standard correlational analysis:

“Briefly, BETA considers differential methylation status (both hypo- and hyper-methylation) in relation to transcription start sites using weighted scores to infer transcriptional regulation, which is then combined with transcriptomic data for validation. This method generates a regulatory potential score on a gene-by-gene basis as well as an overall p value for a cumulative distribution function (one-tailed Kolmogorov-Smirnov test) that discriminates global time point differences for up or down genes.”

Referee #3:

The Authors are to be commended for their thoughtful responses and amendments to the manuscript, that have ultimately strengthened the work. I have no further comments.

We agree with the reviewer that the paper has been significantly strengthened by the collective comments of the reviewers. We want to thank the reviewer for helping us improve the manuscript.

Dr. Kevin Murach
University of Arkansas
303 HPER Building
155 Stadium Drive
Fayetteville, AR 72704
United States

Dear Dr. Murach,

I am very pleased to accept your manuscript for publication in the next available issue of EMBO reports. Thank you for your contribution to our journal.
